# Microscale geometrical modulation of PIEZO1 mediated mechanosensing through cytoskeletal redistribution

Haoqing Jerry Wang [1,2,3], Yao Wang[1], Seyed Sajad Mirjavadi[4], Tomas Andersen[1], Laura Moldovan[1,2,3], Parham Vatankhah[1], Blake Russell[1], Jasmine Jin[1], Zijing Zhou[5], Qing Li[4], Charles D. Cox[5,6], Qian Peter Su [3,7] ✉ & Lining Arnold Ju [1,2,3,8] ✉

The microgeometry of the cellular microenvironment profoundly impacts cellular behaviors, yet the link between it and the ubiquitously expressed mechanosensitive ion channel PIEZO1 remains unclear. Herein, we describe a fluorescent micropipette aspiration assay that allows for simultaneous visualization of intracellular calcium dynamics and cytoskeletal architecture in real-time, under varied micropipette geometries. By integrating elastic shell finite element analysis with fluorescent lifetime imaging microscopy and employing PIEZO1-specific transgenic red blood cells and HEK cell lines, we demonstrate a direct correlation between the microscale geometry of aspiration and PIEZO1-mediated calcium signaling. We reveal that increased micropipette tip angles and physical constrictions lead to a significant reorganization of F-actin, accumulation at the aspirated cell neck, and subsequently amplify the tension stress at the dome of the cell to induce more PIEZO1's activity. Disruption of the F-actin network or inhibition of its mobility leads to a notable decline in PIEZO1 mediated calcium influx, underscoring its critical role in cellular mechanosensing amidst geometrical constraints.

As cells interact with their surrounding microenvironment, they are constantly navigating through complex terrain and microscale geometries shaped by interfaced structures and materials[1–3]. This mechanical microenvironment not only influences cell behavior such as adhesion[4,5], migration[6,7], proliferation[8,9], and differentiation[9–12], but also shapes the subcellular distribution of mechanosensitive proteins, influencing their functionality. Among the key force sensors in this intricate interaction is PIEZO1, a mechanosensitive ion channel ubiquitous across various cell types, which is emerging as a pivotal component of cellular mechanosensing, influencing cell division[13], migration[14], and wound healing[15].

PIEZO1's functionality is intricately linked to its unique structure, which comprises a trimeric propeller-like assembly and 38 transmembrane helices per monomer. This assembly confers a pronounced curvature to the plasma membrane[16], establishing PIEZO1 as a significant modulator of membrane geometry. The inherent curvature of PIEZO1 trimers conveys sensitivity to tension, further influenced by their distinct architecture and size. PIEZO1's role in mechanosensing is highlighted by its preferential localization within the biconcave dimples of red blood cells, crucial for erythrocyte function[17]. Despite considerable progress in elucidating PIEZO family protein structures[18–20] and physiological implications[13,21–23], the biomechanical

[1]School of Biomedical Engineering, The University of Sydney, Darlington, NSW 2008, Australia. [2]Charles Perkins Centre, The University of Sydney, Camperdown, NSW 2006, Australia. [3]Heart Research Institute, Camperdown, Newtown, NSW 2042, Australia. [4]School of Aerospace, Mechanical and Mechatronic Engineering, The University of Sydney, Darlington, NSW 2008, Australia. [5]Molecular Cardiology and Biophysics Division, Victor Chang Cardiac Research Institute, Sydney, NSW 2010, Australia. [6]Faculty of Medicine, St. Vincent's Clinical School, University of New South Wale, Sydney, NSW 2010, Australia. [7]School of Biomedical Engineering, University of Technology Sydney, Sydney, NSW 2007, Australia. [8]The University of Sydney Nano Institute (Sydney Nano), The University of Sydney, Camperdown, NSW 2006, Australia. ✉e-mail: qian.su@uts.edu.au; arnold.ju@sydney.edu.au

modulation of PIEZO1's mechanosensitivity remains less understood. The question of how PIEZO1 interacts with cytoskeletal components, particularly the actin-based cytoskeleton, in the face of microscale geometric modulation is yet to be fully resolved. This interplay gains significance in light of recent studies that indicate a potential functional interaction between PIEZO1 and the cytoskeleton[24] and suggest that the adhesion between cortical cytoskeleton components and the lipid bilayer modifies the membrane tension propagation, potentially influencing PIEZO1 activation[25,26]. Intriguingly, the subcellular localization of PIEZO1, despite its importance, does not follow a uniform pattern. For instance, it is observed that highly curved membrane protrusions like filopodia seem to lack PIEZO1 channels[27], hinting at a potential curvature mismatch preventing PIEZO1 incorporation into these structures[17,28]. Thus, the principles governing PIEZO1 distribution appear complex, potentially involving multiple geometric factors and cytoskeletal interplay.

To bridge this gap, we utilized an ultrasensitive biomechanical approach—fluorescent micropipette aspiration (fMPA) assay[29], that allows us to visualize intracellular calcium mobilization and cytoskeletal redistribution in living cells while simultaneously manipulating the aspiration mechanics and physical microgeometry, including parameters such as narrow space diameter and inclined angle (Fig. 1). By further employing the PIEZO1 transgenic constructs in red blood cell (RBC) conditional knockout mice[23] and HEK293T cells[25,30,31], our study is able to decode how microgeometry influences PIEZO1-mediated mechanosensing and the dynamics of F-actin, a significant constituent of the actin-based cytoskeleton.

## Results

### Microscale geometrical modulation of cell mechanosensing

Our research aims to unravel the intricate relationship between the microgeometry of the micropipette's tip, specifically its diameter and incline (termed "tip angle"), and PIEZO1-mediated calcium ion ($Ca^{2+}$) mobilization, along with F-actin structural redistribution (Fig. 1a). Deepening our understanding of this interaction may inspire innovative therapeutic strategies targeting mechanosensing defects, facilitate the design of biomaterials for tissue engineering applications, and open new horizons for future developments at the human-machine interface. To this end, we focus on two crucial parameters: the micropipette tip angle ($\theta$) and the diameter of the orifice opening ($d$) (Fig. 1b).

Equipped with a high-speed pneumatic pump and a water reservoir, our micropipette holder facilitated precise, steady, and controlled aspiration of targeted RBCs under aspiration pressures $\Delta p = -5$ to $-40$ mmHg[29]. Before aspiration, we pre-incubated cells with the intensiometric calcium dye, Cal-520 AM (see Methods). During the aspiration process, transmitted and fluorescence channels were simultaneously imaged to monitor cellular deformation and quantify $Ca^{2+}$ intensity, respectively (Fig. 1c).

We developed a novel technique to consistently fabricate micropipettes with different geometries, allowing us to examine the influence of these parameters on RBC mechanosensing (Fig. 1c, d). These micropipettes, with either parallel ($\theta = 0°$) or conical ($\theta = 5°$, $10°$) openings, had orifice diameters of $d = 0.94 \pm 0.02$, $1.45 \pm 0.05$, and $2.10 \pm 0.07$ μm (Fig. S1). Under uniform aspiration pressure ($\Delta p = -25$ mmHg), we observed that larger tip diameters led to a noticeable increase in $Ca^{2+}$ intensity. Intriguingly, an increase in tip angle significantly elevated $Ca^{2+}$ influx by 3-fold enhancement relative to the resting state (Fig. 1e; compare $\theta = 0°$ vs. $10°$ at $d = 1$ μm).

### Increasing micropipette tip angle amplifies PIEZO1-mediated $Ca^{2+}$ influx at the cellular tongue over the body

To further elucidate the intricate effects of geometrical parameters on $Ca^{2+}$ influx, we aspirated human RBCs using micropipettes of varying specifications (Fig. 2a, c). The normalized $Ca^{2+}$ intensity change $\Delta F_{max}$ was defined as the Cal-520 AM fluorescence intensity fold change

relative to the resting state[32]. As anticipated, an upward trend in the $\Delta F_{max}$–$\Delta p$ curve was observed when tip diameter increased from $d = 1$ to $2$ μm (Fig. 2a). Interestingly, this pattern was more pronounced when the tip angle of the micropipettes increased from $\theta = 0°$ to $10°$ (Fig. 2a–c). Moreover, the greatest increase in $Ca^{2+}$ influx at certain pressure was observed when both $\theta$ and $d$ of the micropipette were the largest (Fig. 2d). However, the impact of the tip angle ($\theta$) change on the $Ca^{2+}$ influx was more significant than the tip diameter ($d$) change. Indeed, the $Ca^{2+}$ intensity increased from $4.2 \pm 0.3$ to $13.8 \pm 1.0$ fold relative to the resting state when $\theta$ increased from $0°$ to $10°$ (when $d = 1$ μm); whereas an increase in $d$ from 1 to 2 μm only resulted in a slight enhancement from $13.8 \pm 1.0$ to $19.3 \pm 2.0$ fold in the $Ca^{2+}$ intensity (when $\theta = 10°$).

To confirm that this trend was driven by PIEZO1-mediated $Ca^{2+}$ influx, we subjected cells to treatment with Yoda1, Gadolinium ($Gd^{3+}$), and GsMTx4 before the aspiration (Fig. S2; micropipette $d = 1$ μm). As expected, Yoda1 ($0.5$ μM, Fig. S2a), a known PIEZO1 agonist that lowers the mechanical threshold for channel gating[33], shifted the response curve upwards, inducing a significant increase in $Ca^{2+}$ intensity even at -15 mmHg ($\theta = 0°$: $5.06 \pm 0.84$ fold; $\theta = 5°$: $7.57 \pm 1.38$ fold; $\theta = 10°$: $7.14 \pm 0.76$ fold). Conversely, PIEZO1 antagonists $Gd^{3+}$ ($50$ μM, Fig. S2b) and GsMTx4 ($2.5$ μM, Fig. S2c) considerably suppressed RBC mechanosensing across the entire pressure range. Additional experiments with $Piezo1$-KO$^{RBC}$ (see Methods; Fig. S3) provided compelling evidence for the role of PIEZO1 in mediating microgeometry-associated cell mechanosensing.

As the micropipette tip angle increased, we observed distinct $Ca^{2+}$ patterns between the aspirated cell body (portion outside the micropipette) and tongue (portion inside the micropipette) (Fig. 2e). Notably, the tongue exhibited much quicker $Ca^{2+}$ influx than the body when aspirated by a micropipette with a large tip angle (Fig. 2f, $\theta = 10°$). In contrast, when a cell was aspirated by a parallel micropipette (Fig. 2g, $\theta = 0°$), the body experienced a stronger $Ca^{2+}$ influx than the tongue throughout the aspiration process. To quantify such difference, the ratio between the maximum $Ca^{2+}$ intensity of the tongue ($F_{tongue,max}$) and the maximum intensity of the body ($F_{body,max}$) was calculated (Fig. 2h). The results showed that as anticipated, $Ca^{2+}$ influx gradually transitioned from the head to the tongue as the tip angle increased from $0°$ (ratio $= 0.81 \pm 0.16$) to $10°$ (ratio $= 1.41 \pm 0.59$). Importantly, the ratio at a specific tip angle was consistent across different aspiration pressures (Fig. S4). These findings suggest that variations in microgeometry parameters like tip angle can induce differential membrane tension profiles along the cell symmetrical axis, resulting in distinct spatial PIEZO1 activation.

### Microgeometry influences local membrane tension and PIEZO1 activation at the cell-micropipette interface

To discern the effect of microscale geometrical variations on PIEZO1 channel activity, we employed fluorescence lifetime imaging microscopy (FLIM) to examine the membrane tension of RBCs by using tension-sensitive probe, Flipper-TR®[34] (Fig. 3a–c). This tension imaging probe responds to the push-and-pull effect in the lipid bilayer when the membrane experiences tension, and then the probes planarize to exhibit increased fluorescence lifetime $\tau$ when the tension elevates. Thus, an increased fluorescence lifetime was measured in the aspirated cell (Fig. 3d). It is important to note that when the tip angle increases, the Flipper-TR® lifetime increases in both body (Fig. 3d; $\tau = 4.7 \pm 0.1$ ns at $\theta = 0°$, $\tau = 5.4 \pm 0.1$ ns at $\theta = 5°$ and $\tau = 5.4 \pm 0.1$ ns at $\theta = 10°$) and tongue ($\tau = 4.8 \pm 0.1$ ns at $\theta = 0°$, $\tau = 5.6 \pm 0.2$ ns at $\theta = 5°$ and $\tau = 5.9 \pm 0.1$ ns at $\theta = 10°$) regions. Since there is a linear correlation between the fluorescence lifetime and the membrane tension[34], our results provide clear evidence that increasing the tip angle would rapidly increase the membrane tension. As quantified by the Pearson's test, the FLIM data showed a strong positive correlation with the fMPA results (Fig. S5a; $r = 0.83$, $p$-value $= 0.0429$), providing strong evidence that increasing micropipette tip angle increases the membrane

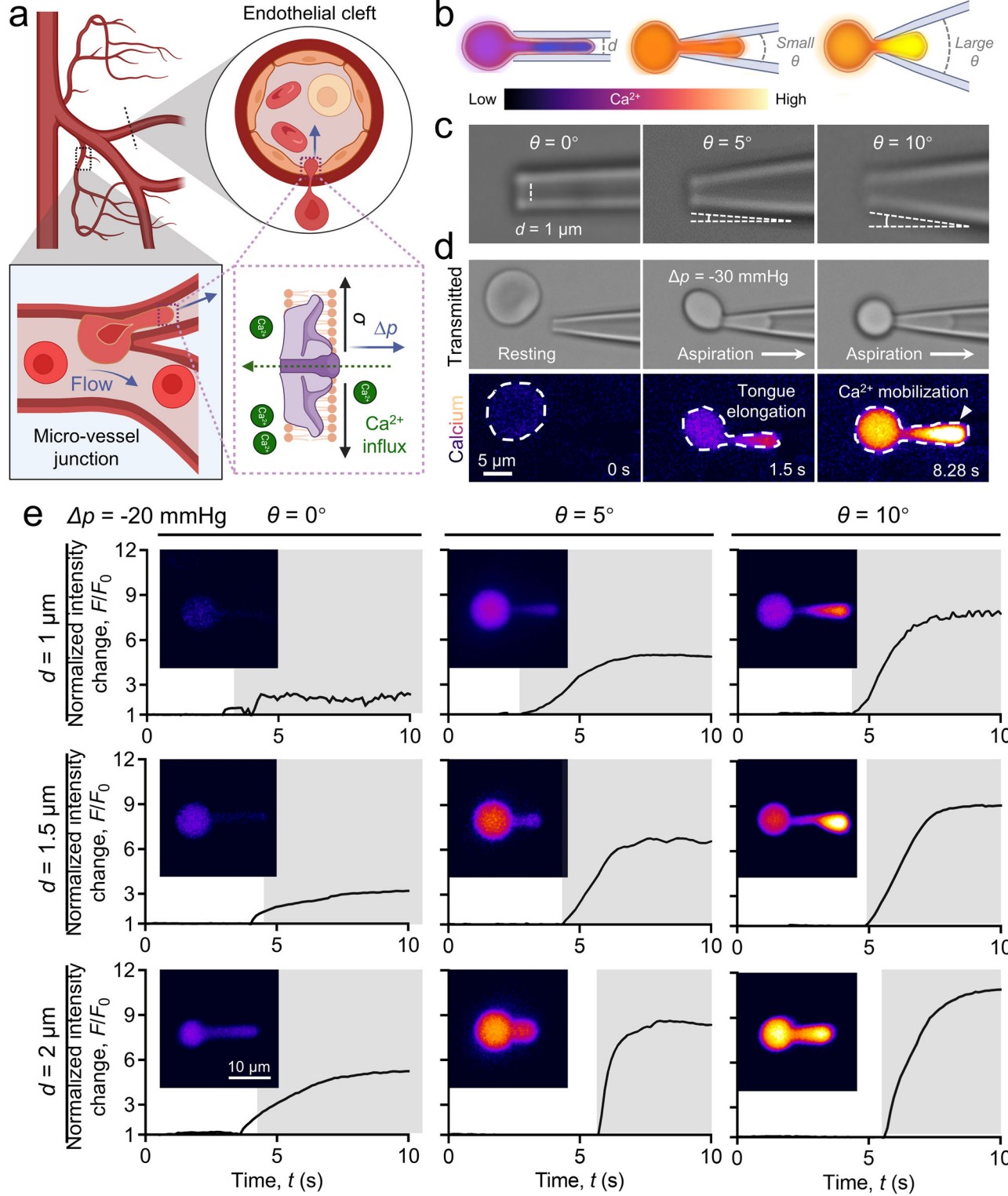

**Fig. 1 | Microgeometry influences mechanically induced Ca²⁺ signaling.**
**a** Schematic of a RBC passing through microenvironments with different geometrical features. When the cell passes through narrow spaces such as micro-vessel junction and endothelial clefts, pressure will induce membrane tension elevation. Mechanosensitive ion channels PIEZO1 (*purple*) on the cell membrane are then activated to allow Ca²⁺ influx. **b** Schematic illustration of the micropipette fabrication with different geometries which finely tuned the tip angle ($\theta$) and diameter ($d$). **c** Representative snapshots of micropipettes with tip angle $\theta = 0°$, 5°, and 10°. All three micropipettes had comparable tip diameter $d = 1$ μm. **d** Brightfield (*top*) and fluorescent (*bottom*) snapshots of the human red blood cell (RBC) being aspirated by the micropipette. When the tongue of the RBC is elongated during aspiration, a significant Ca²⁺ mobilization is observed by the increase in Ca²⁺ intensity in the fluorescent channel. **e** Representative traces of the RBC being aspirated by different micropipettes at $\Delta p = -20$ mmHg. The fold change of Ca²⁺ intensity $F/F_0$ was utilized to examine the Ca²⁺ mobilization inside the aspirated RBC. To this end, the Ca²⁺ intensity increase was enhanced with both increasing tip diameter $d$ and angle $\theta$. **a** and **b** are created with BioRender.com released under a Creative Commons Attribution-NonCommercial-NoDerivs 4.0 International license https://creativecommons.org/licenses/by-nc-nd/4.0/deed.en.

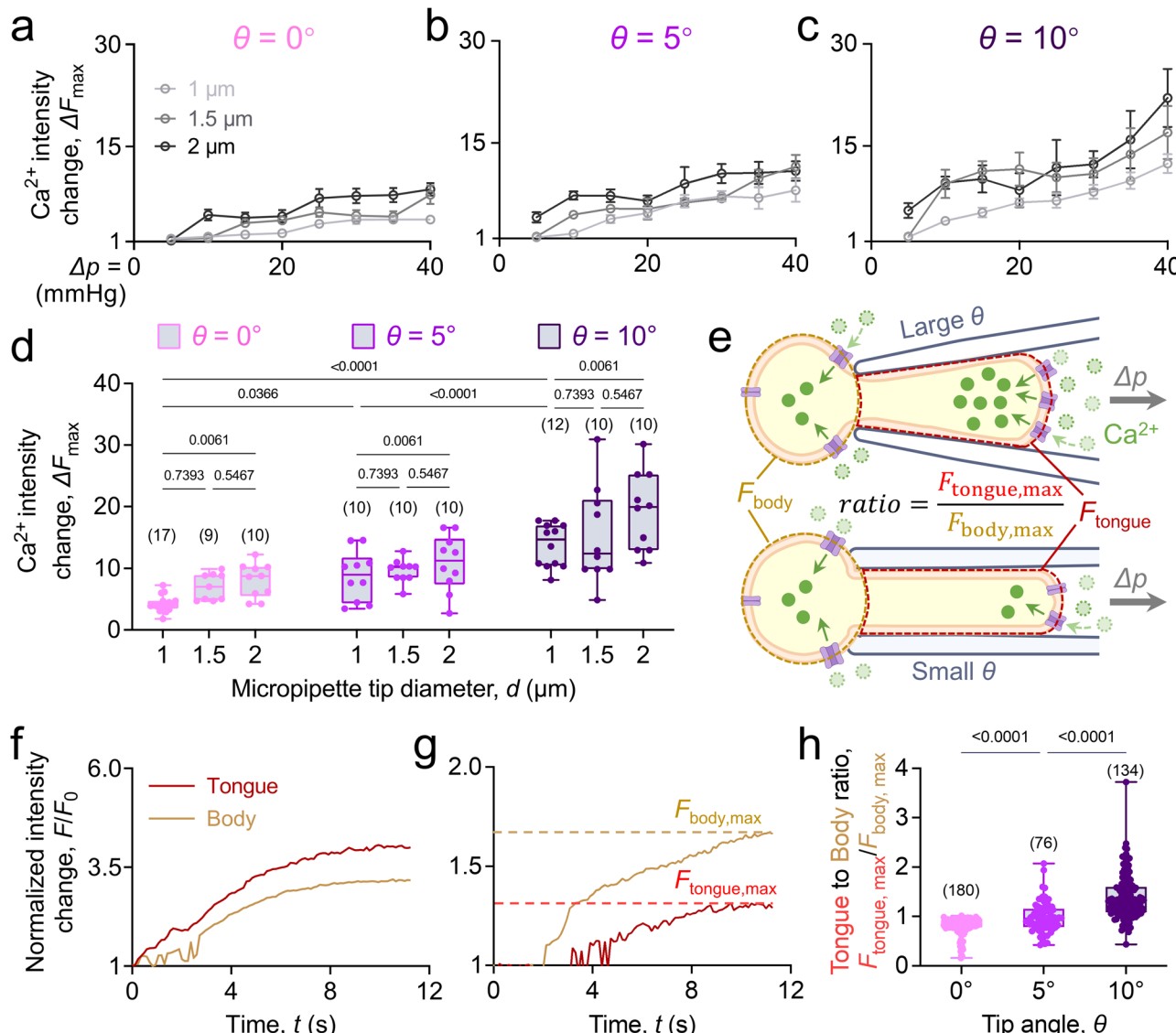

**Fig. 2 | Microgeometry alters PIEZO1 mediated Ca²⁺ mobilization at the tongue of the aspirated RBC. a–c** Human RBCs $\Delta F_{max}$–$\Delta p$ curve under aspiration by micropipette with different geometries. Ca²⁺ fold changes relative to the resting state in the aspirated RBCs were quantified along the pressure ranges from $\Delta p = -5$ to $-40$ mmHg. When the tip angle $\theta$ is fixed, larger diameters upshifted the $\Delta F_{max}$–$\Delta p$ curve, indicating an overall more rapid PIEZO1 activity, while larger tip angle from $\theta = 0°$ (**a**), $5°$ (**b**), and $10°$ (**c**) resulted a stronger upshift effect on the curve. All results were measured from n ≥ 56 cells within 3 independent experiments and presented as mean ± s.e.m. **d** Detailed Ca²⁺ intensity changes comparison amongst different geometries when $\Delta p = -40$ mmHg. Although increasing tip diameter $d$ caused an enhanced Ca²⁺ intensity change due to aspiration, larger tip angle $\theta$ had a more significant enhancement to the Ca²⁺ mobilization. **e** Schematic illustration of the larger tip angle $\theta$ impact on local PIEZO1 activities. We defined the cell part outside the micropipette as the body (*brown circled*) with its Ca²⁺ intensity,

$F_{body}$, while the cell part being aspirated into the micropipette as the tongue (*red circled*), namely $F_{tongue}$. The ratio between $F_{tongue,max}$ and $F_{body,max}$ illustrates the spatial difference of the Ca²⁺ mobilization in the cell. **f, g** Representative normalized Ca²⁺ intensity of $F_{body}$ (*brown*) and $F_{tongue}$ (*red*) when the RBC was aspirated. The $F_{body,max}$ and $F_{tongue,max}$ are indicated by the horizontal dash line in each representative trace, respectively. **h** Ca²⁺ mobilization mapping is neck angle mediated, of which a higher ratio represents more Ca²⁺ influx happens at the tongue of the aspirated cell (*black, $\theta = 10°$*) or a lower ratio represents Ca²⁺ influx was preferrable to happen at the cell body (*light gray, $\theta = 0°$*). The number of cells (*n* value) analyzed in each group was indicated above the bars (**d**, **h**). Data are presented as box plots with medium, minima, and maxima and analyzed by Welch's ANOVA. **e** is created with BioRender.com released under a Creative Commons Attribution-NonCommercial-NoDerivs 4.0 International license https://creativecommons.org/licenses/by-nc-nd/4.0/deed.en.

tension, especially in the tongue, to promote the PIEZO1 mediated Ca²⁺ influx.

Intriguingly, the FLIM results also demonstrate that the membrane tension in the body is slightly higher than the tongue when the cell is aspirated by a $\theta = 0°$ micropipette (Fig. 3e; ratio = 0.98 ± 0.01). This tension distribution shifted when the tip angle increased (ratio = 1.08 ± 0.01 at $\theta = 5°$), and eventually, the tongue experienced a higher tension when $\theta = 10°$ (ratio = 1.14 ± 0.01).

To corroborate our experimental measurements, we employed an elastic shell finite element analysis (FEA) techniques to model the local

membrane tension, as indicated by the maximum principal stress[35], in RBCs subjected to aspiration by micropipettes of varying tip angles (Fig. 3f–h). Our observations showed a significant increase in membrane tension at the tongue of the cell during aspiration (Fig. 3g–h, *black arrow*). These results support the experimental observation that variation in geometry, especially an increase in the tip angle, can promote the activity of PIEZO1 across the cell membrane being aspirated into the micropipette. Interestingly, we also noticed that the highest tension region in RBC aspirated by 0° micropipette was in the middle region of the tongue. This potentially explained the lower Ca²⁺

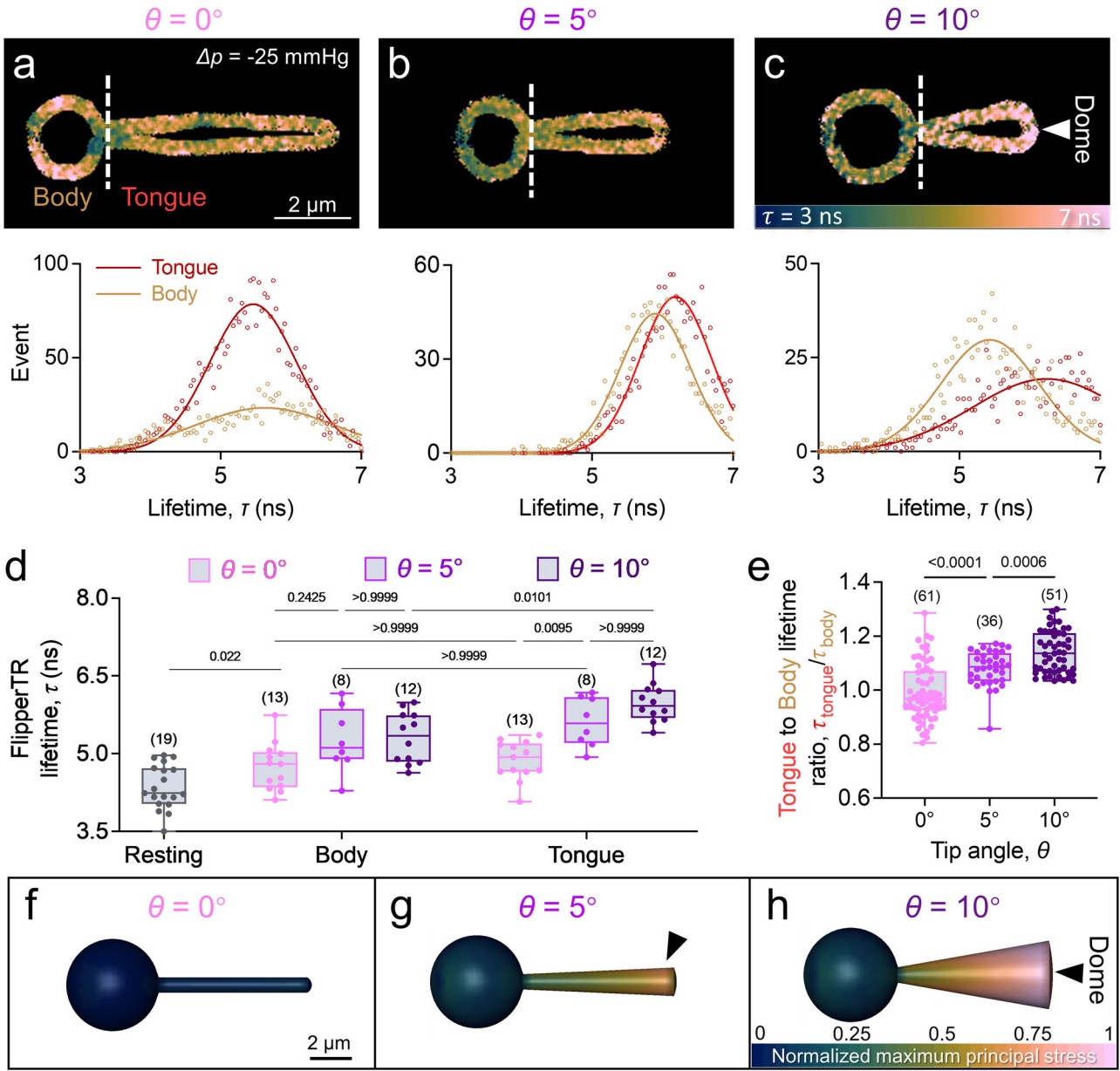

**Fig. 3 | Micropipette tip angle modulates membrane tension at the aspirated RBC tongue. a–c** *Top*: Representative FLIM snapshots using Flipper-TR® reveal tension variations at different aspiration angles. *Below*: Fluorescence lifetime distribution of aspirated cell body (brown) and tongue (red) were plotted. The white arrow points to the cell dome region which has the highest tension when aspirated by $\theta = 10°$ micropipette. **d** Quantitative analysis of Flipper-TR® lifetime changes correspond with altered tension. **e** Tongue-to-body lifetime ratio on RBC aspirated by micropipettes with different tip angles. The number of data points ($n$ number) is

marked above the box and data (**d**, **e**) are presented as box plots with medium, maxima, and minima, and analyzed by Welch's ANOVA test. **f–h** Front view of the FEA simulated aspirated RBC maximum principal stress contours, denoted the change of membrane tension distribution on the aspirated RBCs. Values are normalized based on the maximum principal stress value of all conditions. Black arrow pointed to the regions with the highest membrane tension when the cell was aspirated by $\theta = 0°$ (**f**), 5°(**g**) and 10°(**h**).

influx in the 0° micropipette aspirated RBCs, of which the highest tension was close contact with the borosilicate glass wall. More importantly, the change of tension in the tongue evaluated by the FEA showed a positive correlation with the FLIM results (Fig. S5b; $r = 0.79$) and further confirms our finding that increasing the tip angle would facilitate a much stronger tension elevation in the aspirated tongue, especially in the aspirated dome (Fig. 3c, *white arrow*; Fig. 3h, *black arrow*), which predominantly promotes PIEZO1 activity in the aspirated tongue.

Additionally, we noticed that RBCs aspirated by an angled micropipette demonstrated a distinctive dumbbell morphology at the late stage of fMPA, accompanied by a rapid redistribution of F-actin

(Fig. S6a, b). This phenomenon was more common in RBCs expressing PIEZO1 (Fig. S6c), particularly when aspirated by large tip angle $\theta = 10°$ micropipettes (Fig. S6d). These findings prompt a hypothesis that increased tension at the cell neck due to aspirational forces might drive the redistribution of F-actin, thereby enhancing $Ca^{2+}$ influx within the cellular tongue and amplifying the cell mechanosensing.

**Large tip angle triggers F-actin accumulation as a physical constriction at the aspirated cell neck**
To explore whether the aspiration force can induce F-actin distribution in the aspirated cells, we chose a HEK293T cell line with three PIEZO1 genotypes—wild type (PIEZO1-WT), knock-out (PIEZO1-KO), and

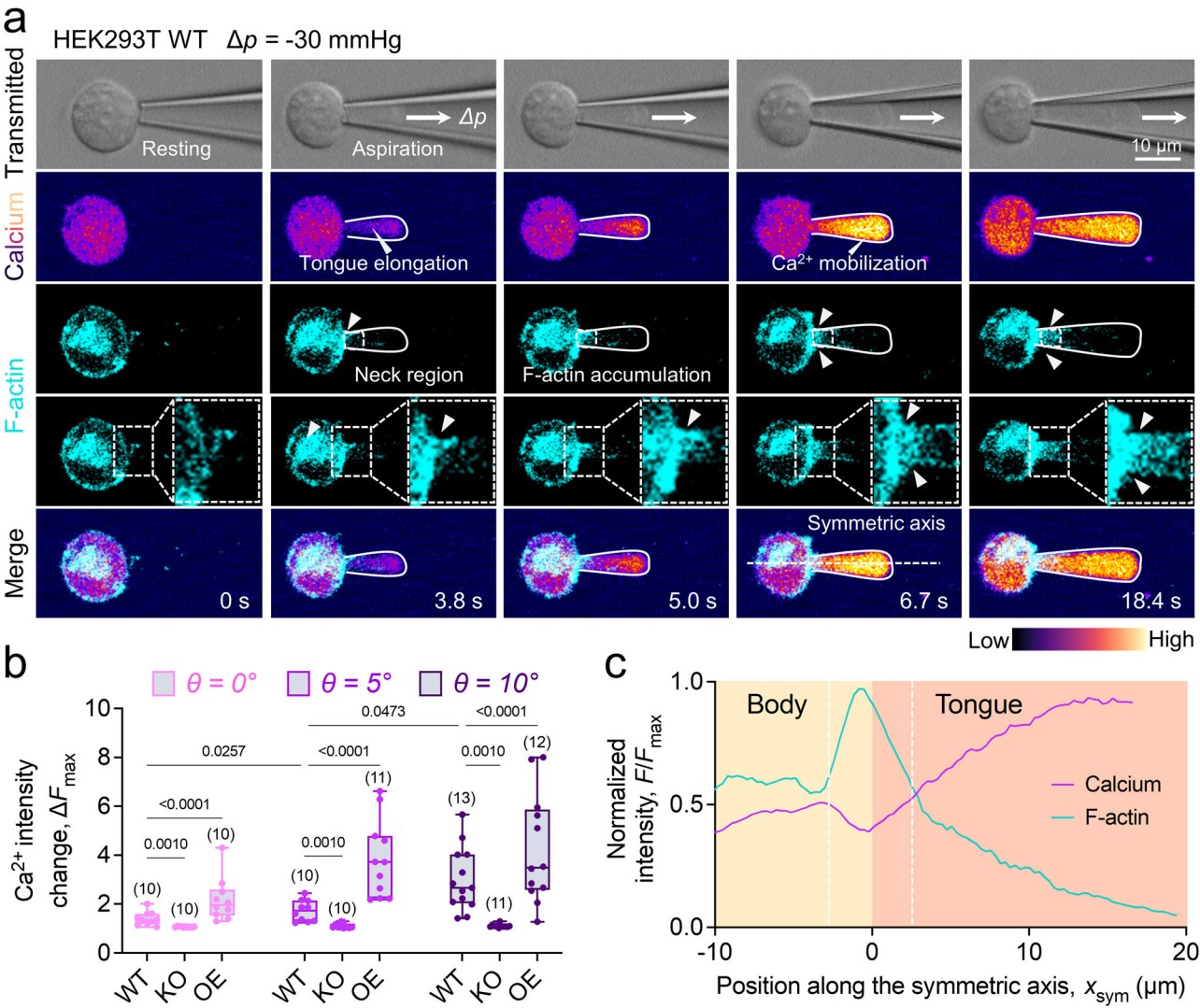

**Fig. 4 | Micropipette tip angle influences the accumulation of F-actin at the cell neck. a** Confocal snapshots of a HEK293T WT cell aspirated by the micropipette. The transmitted channel (*1st row*) was used to visualize the aspiration, while $Ca^{2+}$ mobilization (*2nd row*) and F-actin dynamics (*3rd and 4th row, cyan*) were monitored using a separate scanner unit in the FV3000 microscope, respectively. The zoom in F-actin channel demonstrated the accumulation event at the cell neck region (*dash line* outlined) during aspiration. The merged fluorescence image (*5th row*) showed that F-actin moved to the neck and accumulated during the aspiration, and co-localized with the dividend between low signal cell body and high signal cell

tongue (*t* = 6.7 s). **b** $Ca^{2+}$ fold changes of aspirated HEK293T with different tip angles $\theta$ has a consistent trend with human RBC. The suppressed response in PIEZO1-KO HEK293T (HEK293T KO) and amplified response in PIEZO1-OE HEK293T (HEK293T OE) validated that the $Ca^{2+}$ mobilization in HEK293T cell was PIEZO1 mediated. The number of data points (*n* number) is marked above the box and data are presented as box plots with medium, maxima, and minima, and analyzed by Welch's ANOVA test. **c** F-actin accumulation divided the high PIEZO1 activity tongue from the body. Average intensity profile of calcium (*magenta*) and F-actin (*cyan*) along the symmetric axis was plotted.

overexpression (PIEZO1-OE). We chose HEK293T because it demonstrates superior F-actin mobility compared to RBCs[36,37]. We co-stained intracellular $Ca^{2+}$ and F-actin of each genotype and then aspirated them (Fig. 4a; Fig. S7a−c). As anticipated, $Ca^{2+}$ influx was diminished in the HEK293T KO (Fig. S7b) and amplified in the HEK293T OE (Fig. S7c) under a pressure gradient of $\Delta p = -40$ mmHg and varying tip angles, which substantiated that $Ca^{2+}$ mobilization in the aspirated cell is indeed PIEZO1-dependent (Fig. 4b).

Remarkably, concurrent with these findings, we noticed F-actin accumulated at the aspirated cell neck during aspiration (Fig. 4a, *white arrow*), creating a physical constriction. This accumulation point was even more pronounced in cases where PIEZO1 expression was higher, thus enhancing the probability of F-actin accumulation at the neck (Fig. S8a). Moreover, this F-actin accumulation at the neck region acted as a barrier, segregating the 'high' $Ca^{2+}$ mobilization region within the tongue from the 'low' $Ca^{2+}$ mobilization region in the cell body (Fig. 4c),

elevating the PIEZO1 activity in the tongue which further substantiated our observations in RBCs (Fig. 2h).

### F-actin redistribution forms a physical constriction amplifying PIEZO1 activities

Based on our observations, we postulate that the accumulation of F-actin at the cell neck during aspiration serves as a physical constriction, enhancing the membrane tension at the tongue of the aspirated cell, thereby amplifying the PIEZO1 activity (Fig. 5a). To discern the driving factors behind the F-actin accumulation, we capitalized on the superior F-actin mobility in HEK293T cells. While a higher PIEZO1 expression upregulated the probability of observing accumulation (Fig. S8a) and the time required for F-actin to accumulate at the neck region, $t_{acc}$ (see Methods; Fig. S8b), we also observed $t_{acc}$ conversely decreased with a stronger aspiration (Fig. 5e, when $\theta = 0°$, $t_{acc} = 37.45 \pm 0.74$ s at $\Delta p = -10$ mmHg, $27.02 \pm 0.68$ s at

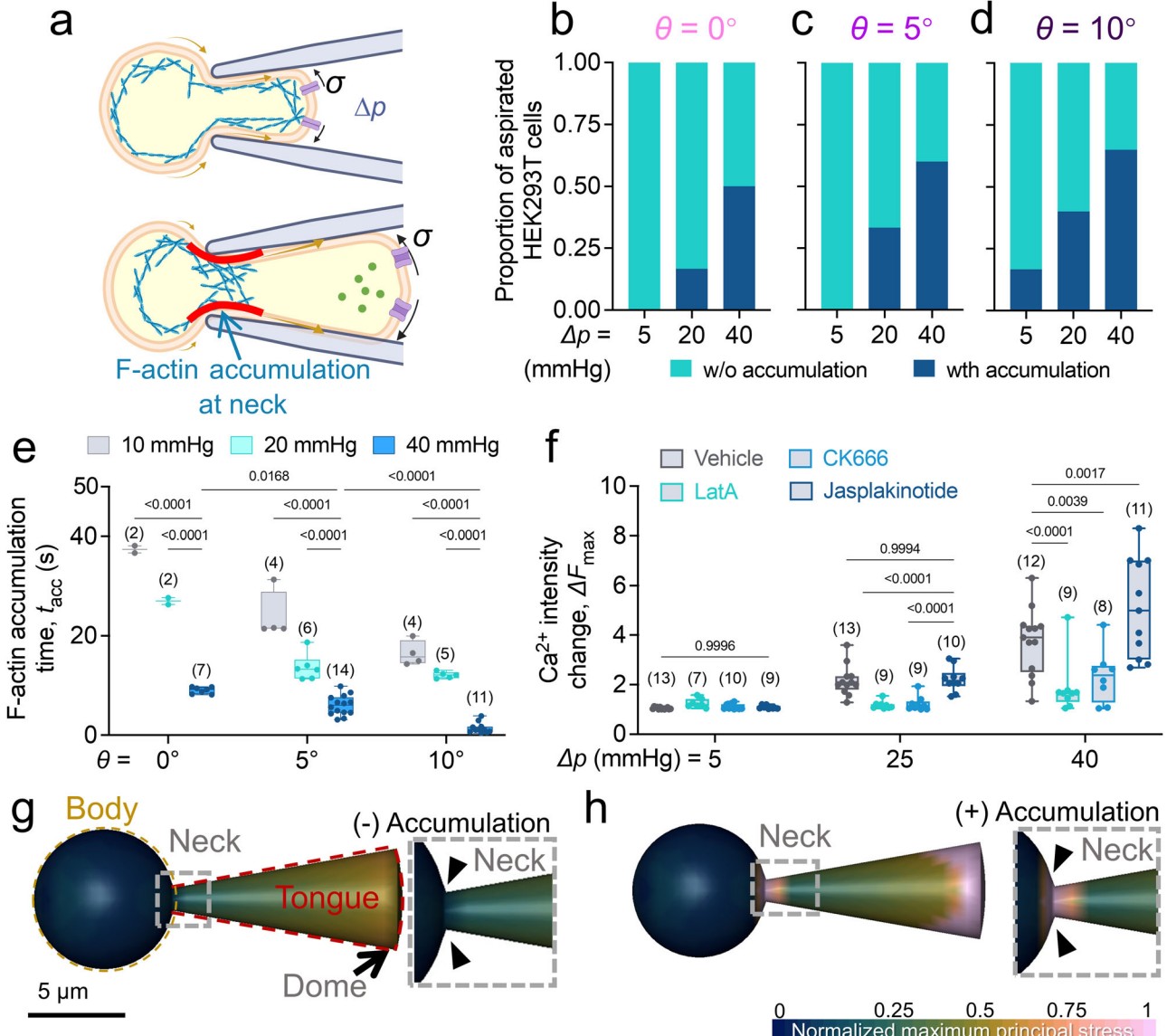

**Fig. 5 | F-actin accumulation influences PIEZO1 activity within the cellular protrusion. a** Schematic illustration of F-actin accumulation effect on PIEZO1 activity at the cell tongue. *Top*: At the early stage of aspiration, F-actin starts to move to the neck from both the body and tongue end of the aspirated cell. *Bottom*: When F-actin accumulates, accumulation impairs the lipid movement due to the aspiration which serves as a physical constriction (*red*) and thereby increases the membrane tension at the cell tongue. As a result, more PIEZO1 activation was expected as the membrane tension increased. **b–d** Proportion of aspirated HEK293T WT cells that exhibited F-actin accumulation at $\Delta p = -5$, $-20$, and $-40$ mmHg. More HEK293T WT cells had F-actin accumulated at the neck when the tip angle increased from $\theta = 0°$ (**d**), to $\theta = 5°$ (**e**) and $\theta = 10°$ (**f**). **e** Interval from the beginning of aspiration for F-actin developed accumulation at the neck. Despite stronger aspiration $\Delta p$, a larger tip angle $\theta$ also boosted the F-actin accumulation. **f** F-actin mesh network modulates the constriction. The HEK293T cells were incubated with either 2.5 μM Latrunculin A (LatA) for 2 hrs, 100 μM CK-666 for 2 hrs to

disrupt F-actin network, or 10 μM Jasplakinolide (Jasp) for 30 mins to further stabilize the F-actin network. Then the cells were aspirated and Ca²⁺ $\Delta F_{max}$ were quantified. Results showed that F-actin network disruption would suppress the Ca²⁺ influx of the cell comparing the DMSO treated (vehicle, *black*) cells at $\Delta p = -25$ and $-40$ mmHg, whereas network stabilization would further enhanced Ca²⁺ influx. The number of data points (n number) is marked above the box (**e**, **f**). Data are presented as box plots with medium, maxima, and minima, and analyzed by Welch's ANOVA test. **g**, **h** Normalized maximal principal stress distribution simulation of an aspirated HEK293T cell. The cell was either no F-actin accumulation (**g**, (-) Accumulation), or with F-actin accumulation at the neck (**h**, (+) Accumulation). Maximum principal stress values were normalized based on the maximum value of both conditions. **a** is created with BioRender.com released under a Creative Commons Attribution-NonCommercial-NoDerivs 4.0 International license https://creativecommons.org/licenses/by-nc-nd/4.0/deed.en.

$\Delta p = -20$ mmHg, and $9.06 \pm 0.25$ s at $\Delta p = -40$ mmHg). Intriguingly, F-actin accumulation was still noticeable even when cells were subjected to aspiration by a parallel micropipette (Fig. 5b, tip angle $\theta = 0°$; Fig. S8c). We uncovered a clear trend that a larger tip angle facilitated a more pronounced F-actin accumulation in the neck region during aspiration (c.f. Figure 5b–d; when $\Delta p = -40$ mmHg, 16.7% of cells has F-actin accumulation aspirated by $\theta = 0°$ micropipette, 60% of cells has F-actin accumulation aspirated by $\theta = 5°$ micropipette, and 65% of

cells has F-actin accumulation aspirated by $\theta = 10°$ micropipette) in PIEZO1-WT and PIEZO1-OE genotypes (Fig. S8a) whereas absence of the mechanosensitive ion channel in PIEZO1-KO genotype leads to few accumulation events recorded in this study and the trend is not clear. Nevertheless, results suggest that this accumulation did not merely depend on the Ca²⁺ influx[38] but was also a result of the mechanical force exerted during the aspiration. Consequently, an increase in the tip angle expedited the F-actin accumulation (Fig. 5e,

when $\Delta p = -40$ mmHg, $t_{acc} = 9.06 \pm 0.25$ s at $\theta = 0°$, $6.023 \pm 0.54$ s at $\theta = 5°$, and $1.48 \pm 0.32$ s at $\theta = 10°$). Of particular interest was the sequential order of events: F-actin accumulated at the neck region first, followed by a surge in $Ca^{2+}$ influx at the cell tongue (Fig. S8d). This sequence resulted in a significantly stronger $Ca^{2+}$ influx when the cell was aspirated by a larger tip angle (Fig. S8c). Together, these observations suggest that the aspiration force prompts a geometry-dependent redistribution of F-actin, creating an accumulation of F-actin at the cell neck. This physical constriction serves to heighten the activity of PIEZO1 in the cell's aspirated region, highlighting a vital and intricate interplay between cell mechanosensing and microgeometry.

We carried out simulations on aspirated HEK293T cells, introducing additional physical constrictions at the neck (Fig. 5g, h). These additional constrictions bolstered the overall membrane tension at the tongue by ~2 fold relative to the simulation without constriction, validating the observation that restricting the lipid bilayer movement significantly enhances the membrane tension of the aspirated region. Further, we scrutinized the contribution of the F-actin network to PIEZO1 activation by treating HEK293T cells with various drugs that modulate the F-actin network (Fig. 5f). For instance, Latrunculin A (LatA) was employed to inhibit F-actin filament formation[39]. We observed that under relatively low aspiration pressure ($\Delta p = -5$ mmHg), the $\Delta F_{max}$ in $Ca^{2+}$ intensity were comparable between LatA treated (Fig. 5f; $1.252 \pm 0.07$ fold) and DMSO treated HEK293T cells ($1.047 \pm 0.012$ fold). However, a clear disparity emerged under higher aspiration pressures ($\Delta p = -25$mmHg and -40mmHg). While the LatA treated cells $Ca^{2+}$ $\Delta F_{max}$ remained constant ($1.175 \pm 0.05$ fold at -25mmHg and $1.87 \pm 0.37$ fold at -40mmHg), the $Ca^{2+}$ $\Delta F_{max}$ in DMSO treated HEK293T cells significantly amplified ($2.165 \pm 0.18$ fold at -25mmHg and $3.73 \pm 0.37$ fold at -40mmHg), suggesting that the absence of F-actin filaments drastically impairs PIEZO1 activity at the tongue under high aspiration pressures. Moreover, the lack of a significant difference between LatA and DMSO treated cells at low aspiration pressure ($\Delta p = -5$mmHg) reinforces the force-dependent nature of F-actin accumulation.

In keeping with these findings, inhibiting the Arp2/3 complex with CK666 ($2.33 + 0.38$ fold at $-40$ mmHg) yielded comparable outcomes to LatA treatment[40], reinforcing the contribution of a robust F-actin mesh in providing the constricting effect at the neck. Interestingly, when we inhibited actomyosin contractility by incubating cells with 30 µM Y-27632 for 30 mins, only a minor impact was observed during the fMPA assay (Fig. S10), demonstrating the independence of this F-actin accumulation from contractility[41]. To further emphasize the importance of the F-actin mesh in enhancing PIEZO1 activity, we treated the HEK293T WT cells with Jasplakinolide, a polymerization promoter[42]. Remarkably, Jasplakinolide treated cells demonstrated a higher $Ca^{2+}$ influx at $\Delta p = -40$ mmHg ($5.10 \pm 0.60$ fold), reinforcing our hypothesis that increased drag force at the cell tongue promotes F-actin accumulation to form a physical constriction.

## F-actin mobility modulates PIEZO1 activation at aspirated cell tongue

To delve deeper into the role of F-actin motility on $Ca^{2+}$ influx, we restricted F-actin and quantified the resulting $Ca^{2+}$ $\Delta F_{max}$ upon cell aspiration. We performed experiments on HEK293T cells either adhered to a fibronectin (FN)-coated cover glass (Fig. 6a) or in a suspended state (Fig. 6b). After imaging the 3D volume, three distinct regions of interest (ROIs) across the cell at different angles were selected (Fig. 6c, d, *white dash line*). We plotted the intensity profile normalized by the maximum intensity in the ROI $F/F_{max}$ along the distance $x$ (Fig. 6e, f). Interestingly, we observed a dense network of F-actin filaments at the cortex of cells adhered to the FN overnight, supporting strong F-actin signal peaks at the cell periphery (Fig. 6c, d, *arrow pointed*). This contrasts with suspended cells where the F-actin is

spread out across the cytosolic region (Fig. 6d, f). The homogeneity of the F-actin signal across the cytoplasm further reinforced this observation.

Aspiration prompted an expected rise in $Ca^{2+}$ influx in both suspended and adhered cells with increasing aspiration pressure. However, a stark difference in F-actin dynamics was noted between the two conditions (c.f. Figure 4a). With the same aspiration pressure $\Delta p$, the movement of F-actin filaments towards the micropipette tip was hindered in the adhered cells (Fig. 6g), consequently suppressing the PIEZO1-mediated $Ca^{2+}$ influx. This resulted in a consistently stronger $Ca^{2+}$ influx in suspended cells across all pressure ranges compared to adhered cells (Fig. 6h, Fig. S10).

These findings suggested that diminished F-actin mobility, impeding F-actin movement towards the neck, influences PIEZO1 gating activity under high aspiration pressure. To quantify the PIEZO1 activity more accurately, we carried out patch-clamp recordings of PIEZO1 channels and compared the activity on suspended (Fig. 6i, *cyan*) versus adhered (Fig. 6i, *black*) cells[25]. Channel activation in adhered cells showed a rightward shift in the gating curve ($P_{1/2, a} = 27.0 \pm 1.3$ mmHg) compared to suspended ones ($P_{1/2, s} = 18.9 \pm 1.0$ mmHg), indicating a higher sensitivity when the cells were aspirated in the suspended state. These results highlight the pivotal role of F-actin dynamics and network structure in providing a physical constriction at the neck to amplify PIEZO1 activity. In the case of adhered cells, the immobilized F-actin structure dampens PIEZO1 activity, inhibiting the cells' response to mechanical stimulation.

## PIEZO1 movements under aspiration occurs independent from F-actin

To confirm that the lower $Ca^{2+}$ mobilization in adhered compared to suspended HEK293T cells was due to decreased PIEZO1 movement not the micropipette, adhered (Fig. 7a) and suspended (Fig. 7b) hP1-mCherry-1591 HEK293T were aspirated at $\Delta p = -25$ mmHg under all angle micropipettes (i.e., $\theta = 0°$, 5°, and 10°) to monitor the PIEZO1 movement on a spinning disk confocal microscope. PIEZO1 in both adhered and suspended cells responded to the aspiration with high signal clusters (*white arrow*) moving into the micropipette, noted by a right shifting average PIEZO1 intensity overtime along the micropipette symmetric axis (Fig. 7c, *adhered*; Fig. 7d, *suspended*). To verify whether PIEZO1 movement was related to the F-actin movement, we analyzed the co-localization factor[43] between PIEZO1 and F-actin. Results showed that the co-localization significantly increased when the suspended HEK293T cell was aspirated (Fig. 7e, *blue*), indicating that PIEZO1 and F-actin responded and moved into the micropipette at the same time until $t = 5$ s. Over the time, the co-localization dropped after $t > 5$ s. Rather than accumulating with the F-actin at the neck of the micropipette (Fig. 4a), PIEZO1 moves with the lipid bilayer into the micropipette. On the other hand, when the cell was adhered to the FN surface, the co-localization was initially lower compared to the suspended cells, indicating that transformed cell lines do not recruit PIEZO1 into focal adhesions[41]. Meanwhile, when the adhered cell was aspirated, the co-localization factor had minor changes over time, further suggesting that PIEZO1 movement occurs independently of F-actin in response to aspiration. Such results were consistent under aspiration with different $\theta$ micropipettes (Fig. S11). Hence, PIEZO1 movement is solely driven by the aspiration force and has limited interaction with F-actin dynamics in this model.

## Discussion

Our study sheds light on the profound influence of microscale geometrical features on cellular mechanosensing. Utilizing a highly refined micropipette aspiration technique, we have examined a broad spectrum of parameters—diameters ranging from 1 to 2.5 µm, tip angles from 0° to 10°, and aspiration pressures from -5 to -40 mmHg. A

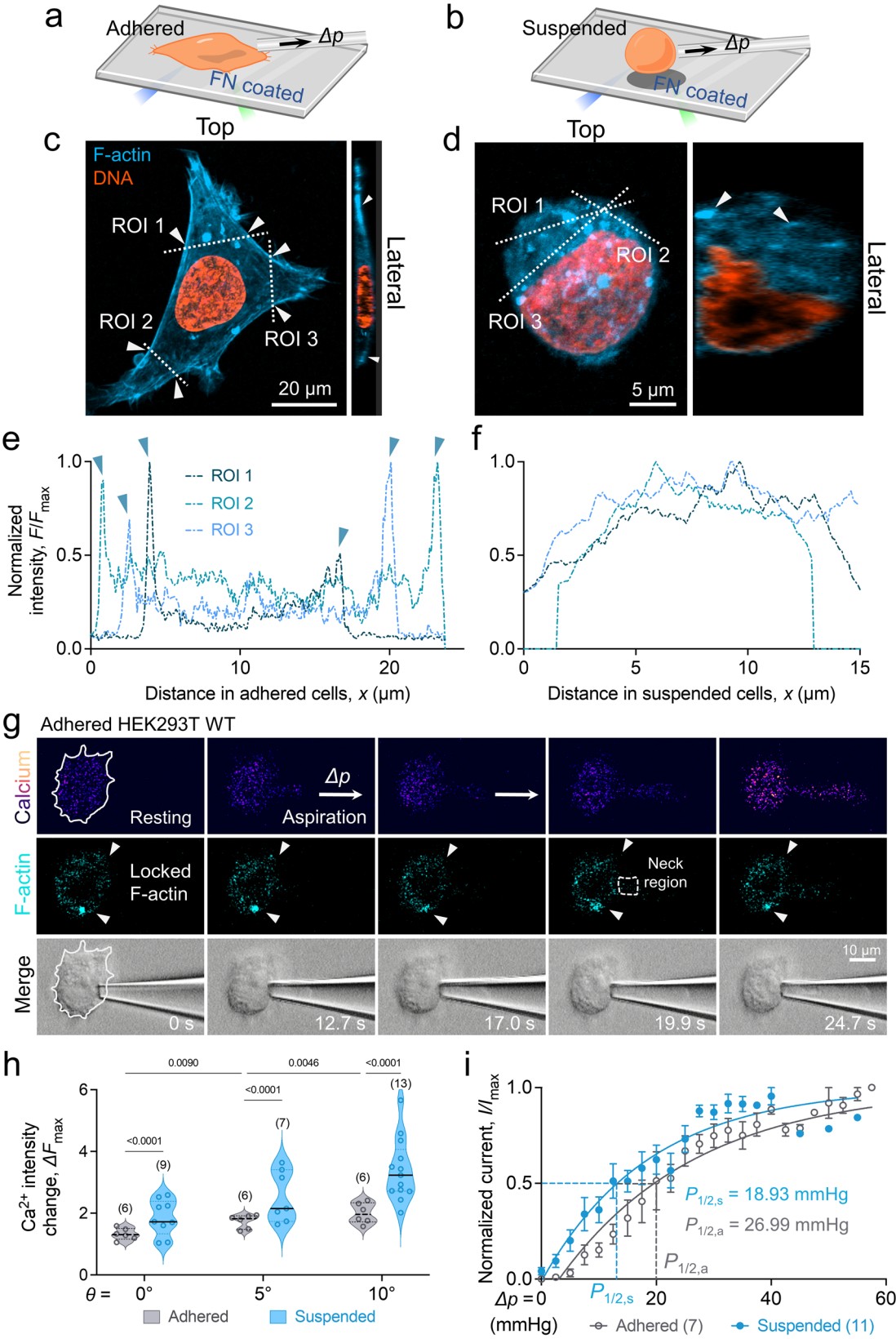

significant discovery from our study is the dynamic role of tip angles on cellular mechanosensing mediated by PIEZO1. As we modulated the tip angles, we observed differing degrees of membrane tension at the aspirated cell region, in turn modulating the mechanosensing response mediated by PIEZO1. Notably, as the tip angle was increased, the mechanosensing response was amplified (Fig. 1e). This discovery

confirms prior assertions about PIEZO1 activity being predominantly modulated by membrane tension[20,25] and adds nuance to our understanding of the variations in Ca[2+] fold change observed in different conditions[29,44].

Additionally, we noticed a spatial variation in Ca[2+] levels in RBCs under higher tip angles. We ascribe this to the heightened PIEZO1

**Fig. 6 | HEK293T PIEZO1 activity is influenced by F-actin mobility. a, b** Schematic illustration of fMPA on adhered (**a**) and suspended (**b**) HEK293T WT. To ensure HEK293T WT cells firmly adhered to the surface during aspiration, the cover glass was coated with fibronectin (FN). **c, d** 3D confocal images of adhered (**c**) and suspended (**d**) HEK293T WT cells. Both lateral and top view of the cells with F-actin (*cyan*) and DNA (*orange*) labeled are shown to illustrate the F-actin distribution in the cell. Three ROI lines were drawn to quantify F-actin signal along the line. **e, f** Normalized F-actin intensity along the ROIs in adhered (**e**) and suspended (**f**) cells. F-actin signal along the line showed that F-actin concentrated at the cell cortex to form organized structure when the cell is adhered to the FN-coated surface, indicated by the signal peaks on either end of the line (*arrow pointed*). **g** Snapshots of aspirated HEK293T cell that was adhered to FN-coated coverglass. The $Ca^{2+}$ and F-actin intensities were concurrently imaged using a FV3000 confocal microscope. The contour of the attached HEK293T cell was outlined (*white dash*

*line*) in the calcium and transmitted channels. White arrows showed that the pointed F-actin signal was immobilized during the whole aspiration process. **h** $Ca^{2+}$ $\Delta F_{max}$ comparison between adhered (*red*) and suspended (*black*) HEK293T WT cells aspirated by micropipette with different tip angles. The number of data points (*n* number) is marked, and data are presented as violin plot, and analyzed by Welch's ANOVA test. **i** Activity of PIEZO1 in suspended and adhered HEK293T cells. A left shift in the Boltzmann distribution was noticed when the cell was suspended comparing to when cells were adhered. Quantified $P_{1/2}$ demonstrated a lower gating pressure was required when cells were suspended. The number of measured cells is marked, and the data is presented as mean ± s.e.m. **a, b** are created with BioRender.com released under a Creative Commons Attribution-NonCommercial-NoDerivs 4.0 International license https://creativecommons.org/licenses/by-nc-nd/4.0/deed.en.

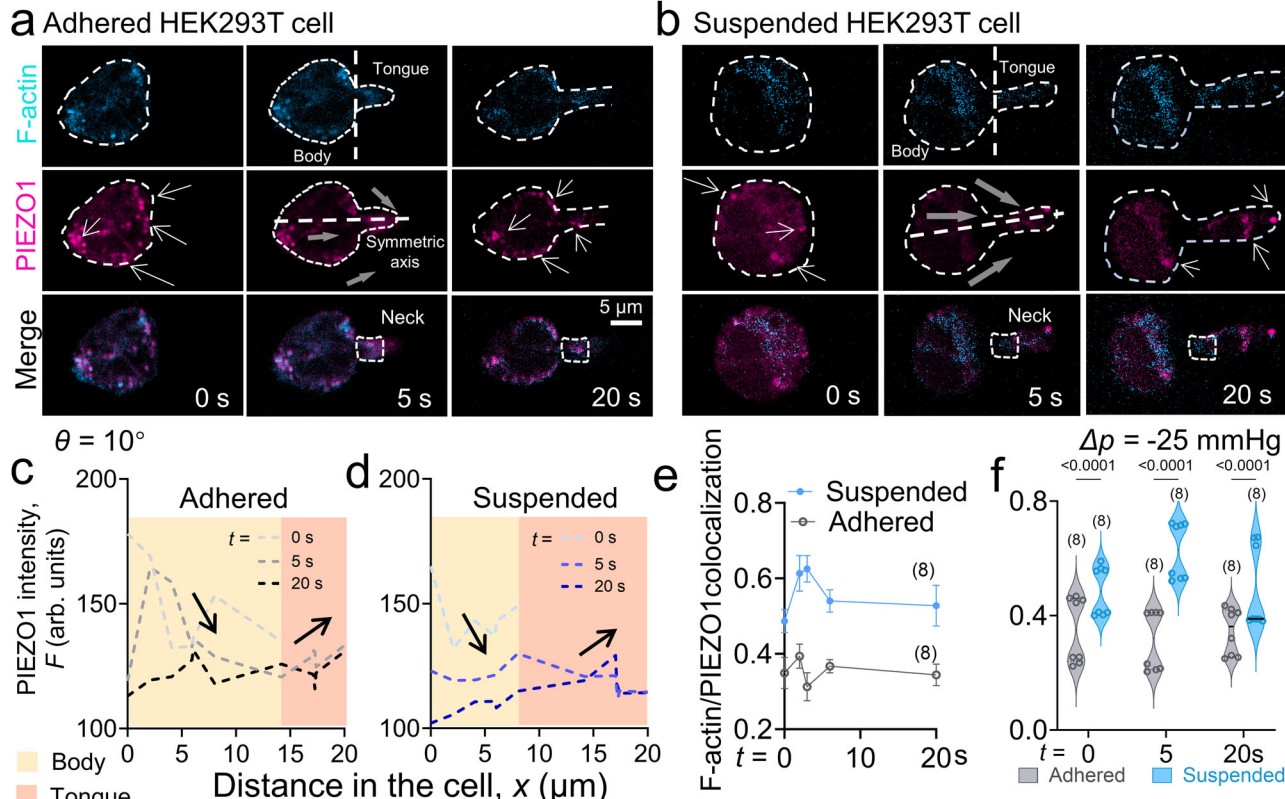

**Fig. 7 | PIEZO1 co-localizes with F-actin during micropipette aspiration in HEK293T cells. a, b** Confocal images of hP1-mCherry−1591 HEK293T cell aspirated at $\Delta p = -25$ mmHg by micropipette ($\theta = 10°$). The F-actin (*cyan, 1st row*) and PIEZO1 (*magenta, 2nd row*) dynamics were monitored during aspiration using a spinning disk confocal microscope. Both adhered (**a**) and suspended (**b**) HEK293T cell showed PIEZO1 clusters (*white arrow*) movement from the cell body to the tongue during aspiration. The symmetric axis was drawn along the aspirating micropipette and across the cell body. Taking the fore-end of the cell body (i.e., left edge) as $x = 0$, mean PIEZO1 intensity was plotted across the adhered HEK293 (**c**) in (**a**) and

suspended one (**d**) in (**b**), respectively. **e** PIEZO and F-actin co-localization were quantified during micropipette aspiration ($\theta = 10°$). Number of cells tested is marked. The data was measured from two independent experiments and presented as mean ± s.e.m. **f** Suspended HEK293T cells had higher co-localization of PIEZO1/F-actin at both resting ($t = 0$ s) and aspirated time points ($t = 5, 20$ s) compared to aspirated HEK293T cells. Images demonstrate a concurrent movement of F-actin and PIEZO1 from the cell body to the cell tongue during aspiration in suspended HEK293T cells. *n* number is marked. Data was presented as violin plots and analyzed by Welch's *t* test.

activity at the tongue of the aspirated cell, which results in the heterogeneity in $Ca^{2+}$ signaling under different geometric conditions. Our combined FLIM tension mapping and elastic shell FEA simulations demonstrate that increased membrane tension, particularly around the aspirated cell's neck region, aligns with the observed upsurge in PIEZO1-mediated $Ca^{2+}$ influx. Notably, our improved FEA simulations clearly illustrate that F-actin accumulation at the neck region leads to a significant increase in membrane tension at the neck and dome of the aspirated cell, correlating with the enhanced PIEZO1 activity observed in the fMPA assay (Fig. 4a, c). This supports the hypothesis that microscale geometrical determinants, i.e., micropipette tip angle and

subsequent physical constrictions, are critical in modulating the mechano-gating of the mechanosensitive ion channel PIEZO1. This has profound implications considering the role of downstream $Ca^{2+}$ channel KCa3.1 in reducing RBC volume[44], and consequently aiding RBCs to pass through narrow capillary spaces[45].

Meanwhile, the tip angle's impact was not limited to RBCs, as it was also observed in the mechanosensing response in the HEK293T cell line. A compelling correlation between the rate of $Ca^{2+}$ influx and the magnitude of the tip angle suggests that cells might employ this relationship as a gauge for estimating available space for passage and possible constriction—a significant finding given the role of $Ca^{2+}$ influx in

localized cytoskeletal redistribution and vital cellular processes[38]. Our study utilized two PIEZO1-expressing cell systems with distinct nanoscale membrane curvatures: RBCs with smooth and flat membranes and HEK293T cells with invaginated membranes. Despite these differences in nanoscale geometry and PIEZO1 localization, both cell lines exhibited similar trends in PIEZO1-mediated Ca²⁺ response under microscale geometrical modulation of the aspirating micropipettes. This suggests that the observed phenomenon is largely independent of microscale curvatures. Furthermore, the limited Ca²⁺ influx in HEK293T cells during the initial aspiration phase (Fig. S8d), when invaginations are expected to flatten, indicates that the later PIEZO1 gating in our system is primarily driven by force-from-lipid mechanisms or lateral force-induced nanoscale footprint changes[16,46–49], rather than local microscale membrane curvature alterations[46,48,50]. However, it is important to consider the complex interplay between membrane tension[51], curvature, and PIEZO1 gating in future studies aimed at understanding the effect of microgeometry on single-cell mechanosensing.

Besides the geometrical modulation on the tension, our study exposes the critical role that F-actin plays as a physical constriction to the lipid bilayer membrane during cellular aspiration. Our findings suggest that the redistribution of F-actin in response to mechanical force is not merely passive but is intricately dependent on the force magnitude (Fig. 5b–e). Notably, we observed a concurrent movement of F-actin and PIEZO1 from the cell body to the aspirated cell tongue, with higher co-localization in suspended cells compared to adhered cells (Fig. 7). This redistribution of F-actin, and the subsequent enhanced PIEZO1 activity, may act as a self-preservation mechanism for cells under increased mechanical stress. Our observations imply that the local constriction from F-actin accumulation aids the cell in maintaining membrane integrity under high tension conditions (c.f. Fig. S6a, b), and this F-actin binding to the lipid bilayer may halt membrane tension propagation from the force application site[26]. This could prevent the dissipation of membrane tension in the aspirated tongue and thus allow an exaggerated Ca²⁺ influx in the tongue.

In adhered cells, well-developed focal adhesions might immobilize a proportion of F-actin filaments, prohibiting them from being drawn into the micropipette and thereby suppressing Ca²⁺ influx at the tongue. Such dynamic interplay between F-actin and PIEZO1 activity is likely more pronounced in suspended cells. Our observation of PIEZO1 dynamics during aspiration illustrates that PIEZO1 moves into the micropipette independent from F-actin dynamics. Together with the patch-clamp results, it is indicated that one key response of F-actin to mechanical stretch is the amplification of the activity of mechanosensitive ion channels after accumulation at the neck (Fig. 6i). Previous research has suggested that F-actin polymerization in adhered cells functions as a protective mechanism for PIEZO1 gating[25]. Extending upon this, our findings underscore the importance of the state of the cell and the site of polymerization. F-actin accumulation at the aspirated neck was found to enhance PIEZO1 gating surprisingly (Fig. 8), suggesting two critical roles: 1) providing mechanical support to prevent membrane rupture, and 2) serving as a physical constriction to the lipid bilayer, which is integral in modulating Ca²⁺ influx at the aspirated cell tongue.

Collectively, our study elucidates the critical role of microgeometry in cellular mechanosensing and uncovers the intricate interplay between F-actin redistribution and PIEZO1 activity in governing cellular responses to mechanical stimuli. We believe our research provides a solid foundation for future investigations into cell mechanosensing, migration, and differentiation. Understanding the intertwined relationship between external mechanical manipulation, transmembrane protein response, cytosolic signaling, and organelle response[52] might pave the way for new strategies in mechano-genetic cell engineering[12], the development of Piezo1 inspired biosensing circuits and therapeutic approaches[53,54].

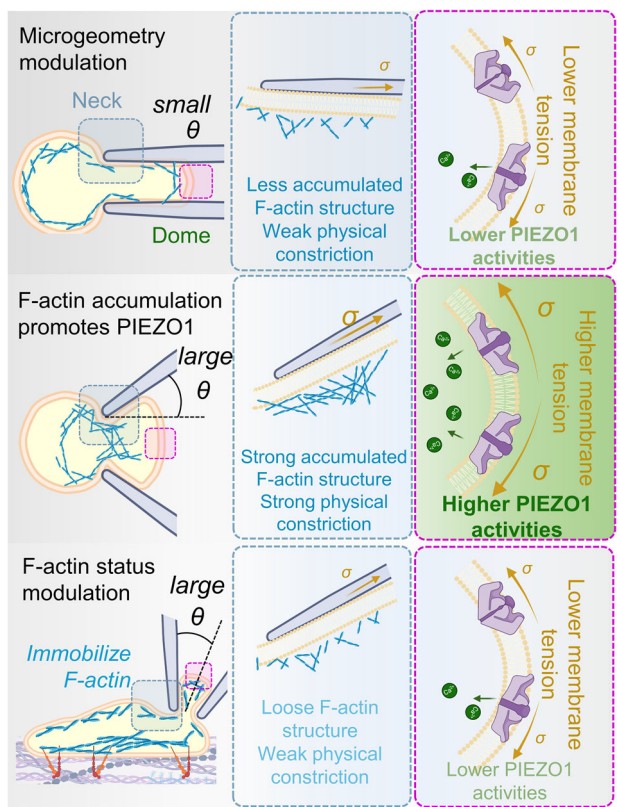

**Fig. 8 | Microgeometry and F-actin interplay during fMPA modulate PIEZO1 activity at the tongue of aspirated cells.** Microgeometry parameters, such as tip angle θ and opening diameter d, globally enhance PIEZO1 activity on the cell membrane while F-actin filaments inside the cell concurrently respond to mechanical aspiration. Reorganized F-actin accumulates at the neck of the aspirated cell, acting as a physical constriction on the membrane to isolate the propagation of membrane tension from the tongue. Concentrated tension at the tongue induces PIEZO1 hyperactivity and results in strong calcium mobilization. Hinderance of F-actin accumulation at the neck, due to either lack of physical constriction (micropipette tip angle θ = 0°) or prohibited actin mobility when cells are adhered to the extracellular matrix, results in low PIEZO1 activity at the tongue. Figure is created with BioRender.com released under a Creative Commons Attribution-NonCommercial-NoDerivs 4.0 International license https://creativecommons.org/licenses/by-nc-nd/4.0/deed.en.

## Methods
### Ethical regulation
This study complies with all relevant ethical regulations. We conducted all human blood collection procedures from healthy volunteers following approval from the University of Sydney's Human Research Ethics Committee (HREC, project 2023/582). All animal experimental procedures were approved by the Animal Ethics Committee of Garvan/St Vincent's (Australia), in accordance with the guidelines of both the Australian Code for the care and use of animals for scientific purposes (8th edition, National Health and Medical Research Council, Australia, 2013). This study did not collect or report the information on sex and gender of the volunteers. No sex-based analysis was included in the study design.

### Ethics statement
All experiments were performed in accordance with relevant guidelines and approved by the University of Sydney Human Research Ethics Committee (HREC, project 2023/582). Informed consents were obtained from human participants of this study. All animal experimental procedures were approved by the Animal Ethics Committee of Garvan/St Vincent's (Australia), in accordance with the guidelines of

both the Australian Code for the care and use of animals for scientific purposes (8th edition, National Health and Medical Research Council, Australia, 2013). All data that support the findings of this study are available upon request.

## Reagents, materials and buffers

Cal-520™ AM (2.5 mM; Abcam, Cambridge, UK; Supplementary Table 1), Yoda1 (1 mM; Cayman Chemical Company, US), Latrunculin A (0.5 mM; Abcam, UK), CK-666 (20 mM; Sigma-Aldrich, US), Jasplakinolide (0.5 mM; Life Technologies, California, US), and Y-27632 (50 mM; Sapphire Bioscience, Australia) was dissolved in anhydrous dimethyl sulfoxide (DMSO, Sigma-Aldrich, US) and stored at −30 °C as per the manufacturer's manual. SPY650-FastAct and Flipper-TR (Spirochrome, Switzerland) were dissolved in 50 µL anhydrous DMSO and stored at −30 °C as per the manufacturer's manual. GsMTx4 (250 µM; Abcam, UK) was dissolved in Mili-Q water and stored at −30 °C as per the manufacturer's manual. GdCl$_3$ (5 mM; Sigma-Aldrich, US) was prepared in Mili-Q and stored at 4 °C as per the manufacturer's manual.

Tyrode's buffer (12 mM NaHCO$_3$, 10 mM HEPES, 0.137 M NaCl, 2.7 mM KCl, 5.5 mM D-glucose, pH 7.2) supplemented with 1 mM CaCl$_2$ was prepared and stored at room temperature. Carbonate/bicarbonate buffer (C-buffer; 2.1 g Na$_2$CO$_3$, 2.65 g NaHCO$_3$ in 250 mL Milli Q H$_2$O, pH 8.5) was stored in 4 °C fridge. Clexane (10,000 U mL$^{-1}$, Sigma-Aldrich, US) was stored at room temperature. A full list of the chemicals and reagents can be found in the Supplementary Materials.

## Micropipette fabrication

A borosilicate glass capillary tube (G-1; Narishige International, US) was mounted on the micropipette puller (model P-1000; Sutter Instruments Co., US). For detailed operation procedures, please refer to the previously published protocols[29,55]. Since the taper length of a conventional angled micropipette is too short to be inserted into our homemade glass chamber[29,56,57], we then developed a multi-stage pulling in the program to fabricate the angled micropipette with enough taper length (Fig. S13; Table 1). To fabricate parallel opening micropipettes ($\theta = 0°$), two close-end micropipettes would then be fabricated by the preset pulling function in the micropipette puller (Heat: 505; Pull: 150; Velocity: 75; Time: 200; Pressure: 400; Ramp 500). Tips of micropipettes were cut and fashioned to diameters of 1–2 µm using a MicroForge (Model MF-900; Narishige International, US) equipped with a 20x eyepiece objective for RBC fMPA assay or 2.5 µm diameter for HEK293T fMPA assay. The pulling programs would result in micropipettes with an opening orifice $d = 1$ µm. Then the pulled micropipettes were cut to the desired orifice with the MicroForge.

## Red Blood Cells (RBCs) collection and labeling

Detailed RBC isolation procedures can be found in our previous studies[29,55,58]. Briefly, we diluted 5 µL of Clexane treated whole blood in 1 mL of C-buffer and centrifuged it at 900 ×g for 1 min at room temperature. We discarded the supernatant and repeated the washing steps with C-buffer and Tyrode's buffer with freshly added 0.5% BSA. We sensitized the PIEZO1 by adding 0.5 µM Yoda1 to the tube before the fMPA. Alternatively, we added 50 µM GdCl$_3$ or 2.5 µM GsMTx4 for 30 min to suppress the PIEZO1 activity[59,60].

Similar to the previously described methods[23,44], we generated RBC-specific PIEZO1 knockout mice (*Piezo1*-KO$^{RBC}$) with erythropoietin receptor (EpoR) Cre recombinase[61]. Subsequently, we introduced the *Piezo1*$^{flox}$ alleles to the EpoR-Cre mice. The first generation of this crossbreeding resulted in all traits being heterozygous. The next step involved back-crossing these heterozygous mice with homozygous *Piezo1*$^{flox}$ animals to produce the *Piezo1*-KO$^{RBC}$ experimental subjects, where the EpoR-Cre induced recombination specifically in the erythroid lineage, knocking out the *Piezo1* gene. All male mice with C57BL/6 J background and carrying and EpoR-Cre were bred and sourced from Australian BioResources (ABR; Moss Vale, NSW, Australia). *Piezo1*$^{flox}$ mice were obtained from Jackson Laboratories (JAX stock 029213). The mice were maintained in a light/dark cycle of 12 h/12 h, at a temperature of 21 °C and 50% humidity.

To collect mouse RBCs, we anaesthetized mice (6 to 8 weeks old, both males and females) with either sodium pentobarbitone (60 mg kg$^{-1}$) or ketamine/xylazine (150/15 mg kg$^{-1}$). We then collected mouse blood via the inferior vena cava as previously described[62], isolated the RBCs, and maintained them at a concentration of $1 \times 10^7$ mL$^{-1}$. For the fMPA assay, we supplemented 150 µL washed RBCs with 1:150 Cal-520 AM (16.67 µM) and incubated at room temperature for 1 hr. To label F-actin, we added 1:1000 SPY650-FastAct to the cell mixture 1 hr before the calcium intensiometric dye. We maintained the concentration of dye probes throughout all experiments.

## Cell culture, harvesting and labeling

We cultured HEK293T cells with PIEZO1 wildtype (WT), knockout (KO; kindly gifted from Dr Ardem Patapoutian)[30], and overexpressing (hP1-mCherry-1591 OE; kindly gifted from Dr Philip Gottlieb)[63] in RPMI Medium 1640 (Thermo Fisher Scientific, US) supplemented with 10% Fetal Bovine Serum (FBS; Thermo Fisher Scientific, US) and 1% Penicillin/Streptomycin (PS; Thermo Fisher Scientific). We maintained cells at 5% CO$_2$ and 37 °C.

Before experiments, we detached cells using TrypLETM Express (Thermo Fisher Scientific, US) and transferred them into a 96-well cell culture plate (Thermo Fisher Scientific, US) at approximately 50% confluency in the RPMI medium. We incubated the cells at 37 °C overnight. For the fMPA on suspended cells, we added 1:000 Cal-520 AM (2.5 µM) and SPY650-FastAct dye to the wells and incubated them in a 37 °C incubator for 1 h. To disrupt the F-actin network, we concurrently treated cells with 2.5 µM LatA for 2 h, 100 µM CK-666 for 2 h, 10 µM Jasp for 30 min, 30 µM Y-27632 for 30 min, or 1:500 DMSO for 2 h at 37 °C. We then harvested the cells using TrypLE Express (Thermo Fisher Scientific) and spun them down at ~200 × g for 5 min. We removed the supernatant and resuspended the cell pellets in Tyrode's buffer supplemented with the same dye and drug concentration.

**Table 1 | Parameter setting in Sutter P−1000 micropipette puller (model P−1000; Sutter Instruments Co., US)**

| Tip angle θ | Stage | Heat | Pull | Velocity | Time | Pressure | Repetition |
|---|---|---|---|---|---|---|---|
| 0° | 1 | Ramp+10 | 150 | 75 | 250 | 500 | once |
| 5° | 1 | Ramp+14 | 0 | 90 | 255 | | once |
| | 2 | Ramp+14 | 0 | 32 | 255 | 670 | twice |
| | 3 | Ramp+4 | 0 | 37 | 255 | | once |
| 10° | 1 | Ramp+14 | 0 | 90 | 255 | | once |
| | 2 | Ramp+14 | 0 | 32 | 255 | 670 | twice |
| | 3 | Ramp+4 | 0 | 29 | 255 | | once |

All parameters are defined based on the application, slight modification is required for every new setup. Ramp value was measured based on the heating filament in the puller.

To perform fMPA on attached cells, we harvested cells from wells and seeded them on a 24×40 mm cover glass coated with 50 μg mL⁻¹ human fibronectin (FN, Thermo Fisher, US). We covered the cover glass with RPMI medium and placed it in a 37 °C incubator overnight to allow the cells to adhere firmly. Before the fMPA, we supplemented the medium with dye probes and incubated it further for 1 hr. We then gently replaced the liquid with Tyrode's buffer.

## Fluorescent micropipette aspiration assay (fMPA)

A detailed protocol for the fMPA assay was outlined in our previous study[29]. Briefly, a prepared glass micropipette was mounted into a micropipette holder (Narishige International, US), which was connected to a high-speed pneumatic pressure clamp device (HSPC; ALA Scientific Instruments, US). Two coverslip slices were adhered to the top and bottom of the chamber holder respectively, with the cell solution introduced between them. The chamber was subsequently positioned on a custom manual stage. By utilizing the μMp-3 Triple Axis Micromanipulator (SENSAPEX, Finland), the micropipette was gently introduced into the chamber from the side at an angle of approximately 10°. The aspiration pressure, ranging from $\Delta p = -5$ to $-40$ mmHg, was controlled using a custom LabVIEW program (National Instrument, US).

For concurrent imaging of the Ca²⁺ signal and transmitted light, a customized light path was integrated into the Olympus IX83 inverted microscope, equipped with a 60× dry objective (NA 0.7). A pE-300 (CoolLED, UK) LED light source served as the fluorescence light source, emitting 488 nm excitation light which was reflected by the first dichroic mirror to excite the calcium dye probes. The 510 nm emission light from Cal-520 was passed through the first dichroic mirror, reflected by the second dichroic mirror (long pass: 560 nm), and captured by the ultrasensitive sCMOS Prime 95B camera (Teledyne Photometrics, US). For the transmitted light path, a 760 nm long-pass filter was placed in front of the light source to facilitate transmission through all DMs, and the resultant images were captured by a Manta GigE G-040 CMOS camera (Allied Vision, Germany). Both cameras were controlled by μManager.

Olympus FV3000 confocal microscope with two high sensitivity detectors (HSD) was utilized to image three channels concurrently (i.e., transmitted, 510 nm fluorescence, and 674 nm fluorescence). The chamber was placed on our designed manual stage integrated with the micropipette manipulator. A 40× silicon oil-immersion objective (NA 1.25) was used while the images were illuminated using the silver-coated resonance scanning mirrors at 10 frames per second (fps). A 1.6× magnification was implemented during acquisition using the Olympus FV31S-SW software.

To monitor PIEZO1 and F-actin dynamic under fMPA, a Nikon Ti2-E Inverted microscope with three Photometrics Prime 95B back illuminated sCMOS cameras for simultaneous imaging was utilized to obtain fast 2D acquisitions in three channels concurrently with minimal photobleaching (i.e., 488 nm (Laser: 350 mW), 560 nm (Laser: 550 mW), and 642 nm (Laser 550 mW) fluorescence). All three channels were acquired with an exposure time of 200 ms. The coverslip was placed on a universal stage integrated with the micropipette manipulator. A 100× silicon oil-immersion objective (NA 1.35) was used while images were acquired using spinning disk confocal microscopy (Pinhole: 50 μm) at 10 fps. A 1× magnification was used during the acquisition utilizing the Slidebook capture software (Intelligent Imaging Innovations, US).

## Fluorescence intensity analysis and F-actin accumulation tracking

The fluorescence intensity of the aspirated cell over time was quantified using Imaris 9.0.1 (Oxford Instruments, UK). Background signals were subtracted from all measurements. The normalized Ca²⁺ intensity change was then calculated using the following equation:

$$\Delta F_{max} = \frac{F_{max}}{F_0} = \frac{F_{max,absolute} - F_b}{F_{0,absolute} - F_b} \tag{1}$$

where $F_{max,absolute}$ is the absolute maximum intensity of the aspirated cell, $F_{0,absolute}$ is the absolute intensity of the cell at the resting state, and $F_b$ is the background intensity.

Taking the tip of the micropipette as the boundary, a small $20 \times 20$ pixels ROI was drawn around the micropipette neck to examine the F-actin accumulation time $t_{acc}$. Aspirated cell's F-actin signal inside the ROI $F_{a,neck}$ is tracked over the whole aspiration process. Then a ratio between the neck and whole cell signal $F_{a,cell}$ was derived:

$$ratio_{F-actin} = \frac{F_{a,neck}}{F_{a,cell}} \tag{2}$$

If the ratio increased more than 50% and was stabilized for more than 1 min, we define the F-actin accumulated. The time taken for F-actin to accumulate, $t_{acc}$ was recorded by measuring the interval between the timepoint that aspiration started, $t_{pressure}$ and the time-point that ratio increased to the threshold, $t_{50\%}$:

$$t_{acc} = t_{50\%} - t_{pressure} \tag{3}$$

## PIEZO1/F-actin co-localization analysis

To track hP1-mCherry movement in HEK293T cells during aspiration, the fluorescence intensity of the aspirated cell over time was quantified using FIJI 2.15.0. Background signals were subtracted from all measurements. An ROI was drawn around the boundary of the cell body and aspirated cell tongue. The fluorescence intensity was then measured along the centroid of the micropipette at three time points, where 0 s was the point immediately prior to aspiration. The co-localization of PIEZO1 (609 nm) and F-actin (674 nm) fluorescence signal of the aspirated cell at baseline and over time was quantified using FIJI 2.15.0 with the JACoP 2.1.1. Background signals were accounted for utilizing the corrected correlation method by Alder et al.[43]. Briefly, a correction factor was derived for each acquisition based on the formula below:

$$c = \frac{1}{\sqrt{R_{560} \times R_{647}}} \tag{4}$$

Where $c$ is the correction factor, $R_{560}$ is the Pearson's correlation for two concurrent time points (0.1 s interval) in the PIEZO1 channel, and $R_{647}$ is the Pearson's correlation for two concurrent time points (0.1 s interval) in the F-actin channel. Further, a mean measured correlation was found by taking two concurrent time points (0.1 s interval) in each channel and obtaining four Pearson's correlation factors for each combination of channels ($560_1/647_1$, $560_1/647_2$, $560_2/647_1$, $560_2/647_2$). This mean measured correlation was then multiplied by the correction factor to obtain the final corrected correlation. This corrected correlation was used to quantify co-localization of PIEZO1 and F-actin over time during aspiration. Co-localization was quantified at five time points, where $t = 0$ s was the point immediately prior to aspiration.

## 3D confocal imaging

To prepare the adhered HEK293T cells for imaging, they were seeded into a fibronectin-coated Lab-Tek™ II 8-well chamber coverglass (Thermo Fisher Scientific, US) at a density of ~ 5,000 cells well⁻¹ and incubated at 37 °C a day prior to the experiment. The cells were then stained with SPY650-FastAct for 1 hr and Hoechst 33342 for 30 mins at

37 °C before the confocal imaging. The Olympus FV3000 confocal microscope (Olympus Corporation, Japan), equipped with 2 High-Speed Disc (HSD) units, was used for multicolor imaging. The coverglass was placed in a stage-top incubator (Tokai Hit, Japan) at 37 °C with 5% $CO_2$ supplementation. To excite Hoechst 33342 and SPY650-FastAct, lasers at 408 nm and 647 nm were used, respectively. The images were scanned at a 1024×1024 (pixel × pixel) resolution and 0.4 µm z-step size.

## FLIM experimental setup and data analysis

For FLIM imaging, human RBCs were washed and resuspended into the Tyrode's buffer without BSA. The cells were then incubated with 1 µM Flipper-TR dye probes for 15 min at RT. Then the cells were further diluted with an equal volume of Tyrode's buffer with 1% BSA and then injected onto the coverslip to perform MPA during FLIM imaging.

The FLIM setup was built on a Nikon A1R confocal microscope with CFI Plan Apo Lambda 60× oil immersion objective (Nikon, Japan) which has a 1.4 numerical aperture and 0.13 mm working distance. The images were scanned in 512×512 (pixel×pixel) resolution with 1/4 frame per second. To ensure enough photon counting for lifetime analysis, temporal resolution was sacrificed and a total number of 5 frames (total acquisition time = 20 s) were stacked for each acquisition. The FLIM lifetime data was acquired with a pulsed femtosecond laser operating at a repetition rate of 40 MHz. A filter cube with 475/28 nm excitation filter, 561 nm dichroic cut-off mirror, and 609/57 nm emission filter was in place to separate fluorescence. The microscope setup is designed for fluorescence lifetime imaging, and the FLIM data was collected by 2 PMA-Hybrid 40 detectors and single-photon timing was determined by the time-correlated single-photon counting (TCSPC) electronics module, TimeHarp 260 Pico TCSPC unit (PicoQuant, Germany). Using the photodetector signal, galvanometer clocks, and pulsed laser sync signals, the photon arrival time is measured, and single-pixel histograms are generated. SymPhoTime 64 software (PicoQuant, Germany) was used to control the system, perform the scanning and record both single-pixel histograms and raw data. The instrument response function (IRF) was extracted from the SymPhoTime 64 software after each experiment.

Fitting of FLIM images from raw data was performed with the FLIMfit software tool (version 5.1.1) developed at Imperial College London[64]. Images were spatially binned 1×1 and a temporal binning of the fluorescence decays was applied prior to fitting, resulting in 250-time bins per decay (100 ps bin⁻¹). Fitting of the fluorescence images was then performed pixel-wise with a double exponential model and using a maximum likelihood estimation algorithm, whereby the longer lifetime $\tau_1$ was extracted. A 3×3 smoothing kernel was defined, allowing spatial variations in fluorescence lifetime to be visualized. Background extraction and segmentation were applied to enhance the quality of the photon counting analysis.

## Elastic finite element analysis on single cell micropipette aspiration

The simulation of micropipette aspiration was conducted using ANSYS LS-DYNA R13.1.0 (ANSYS, US), a commercially available finite element (FE) analysis software. During the initial stage of simulation, the RBC geometry was considered based on a continuous degree 4 surface which is constrained by three principal parameters: the main diameter $d$, the thickness at the dimple centre, $b$, and the maximum thickness, $h$[65]. The geometry is reconstructed based on the equation given by:

$$\left(x^2 + y^2 + z^2\right)^2 + P\left(x^2 + y^2\right) + Qz^2 + R = 0 \qquad (5)$$

where,

$$P = -\frac{d^2}{2} + \frac{h^2}{2}\left(\frac{d^2}{b^2} - 1\right) - \frac{h^2}{2}\left(\frac{d^2}{b^2} - 1\right)\left(1 - \frac{b^2}{h^2}\right)^{\frac{1}{2}} \qquad (6)$$

$$Q = P\frac{d^2}{b^2} + \frac{b^2}{4}\left(\frac{d^4}{b^4} - 1\right) \qquad (7)$$

$$R = -P\frac{d^2}{4} - \frac{d^4}{16} \qquad (8)$$

The geometrical parameters are given as $d = 8$, $b = 1$, and $h = 2.25$ µm (Fig. S14a). The RBC model was firstly generated in SOLIDWORKS 2021 (Dassault Systèmes, France), and exported as an IGS format. The IGS file was then imported to LS-PrePost (ANSYS V4.5.24) for generating mesh and applying boundary and loading conditions. Finally, the finite element problem was solved in LS-Run 2023 (ANSYS, US).

In this study, the RBCs were considered as an elastic shell with a thickness of 200 nm to ensure deformability during simulation (Fig. S14a). To generate the finite element model of RBC, the material was considered isotropic and linear elastic, and thus, the stress-strain relationship can be expressed as:

$$\boldsymbol{\sigma} = \mathbf{C} : \boldsymbol{\varepsilon} \qquad (9)$$

where $\boldsymbol{\sigma}$ and $\boldsymbol{\varepsilon}$ are the stress and strain tensors, respectively[66]. The stress can be also written as:

$$\boldsymbol{\sigma} = \begin{bmatrix} \sigma_{xx} & \sigma_{xy} & \sigma_{xz} \\ \sigma_{xy} & \sigma_{yy} & \sigma_{yz} \\ \sigma_{xz} & \sigma_{zy} & \sigma_{zz} \end{bmatrix} \qquad (10)$$

where $\sigma_{rs}$ ($r,s = x,y,z$) denotes three-dimensional stress components in Cartesian coordinate system. The fourth-order elasticity tenor $\mathbf{C}$ can be simplified as:

$$\mathbf{C} = \frac{E_{\text{aspiration}}}{(1+\upsilon)(1-2\upsilon)} \begin{bmatrix} 1-\upsilon & \upsilon & \upsilon & 0 & 0 & 0 \\ \upsilon & 1-\upsilon & \upsilon & 0 & 0 & 0 \\ \upsilon & \upsilon & 1-\upsilon & 0 & 0 & 0 \\ 0 & 0 & 0 & \frac{1-2\upsilon}{2} & 0 & 0 \\ 0 & 0 & 0 & 0 & \frac{1-2\upsilon}{2} & 0 \\ 0 & 0 & 0 & 0 & 0 & \frac{1-2\upsilon}{2} \end{bmatrix} \qquad (11)$$

where $E_{\text{aspiration}}$ denotes the Young's modulus of RBC shell (8 kPa for $\theta = 0°$; 41.7 kPa for $\theta = 5°$; 41.2 kPa for $\theta = 10°$), $\upsilon$ is the Poisson's ratio (0.49) which were determined by calibrating the relative aspirational pressure ($\Delta p = 25$ mm Hg) exerted onto the dome section.

Based on the experimental observation, the initial and final geometry of the RBC were obtained based on the images we captured from the experiment. The displacement loading was applied onto all the nodes in the original RBC shell. The simulations were performed in LS-Run 2023 (ANSYS, US). In order to compare the results, the maximum first principal stress was obtained for each studied case[35] that can be calculated from,

$$\det(\boldsymbol{\sigma} - \lambda\,\mathbf{I}) = 0 \qquad (12)$$

where $\mathbf{I}$ denotes the identity and $\lambda$ is the eigenvalue. The values reflected the trend of membrane tension elevation due to the aspiration[35] and were normalized based on the maximum value of all conditions at each cell type.

When the F-actin accumulates at the neck region of aspirated HEK293T cell, the material properties are expected to increase significantly[67,68]. In our proposed material model, we introduce an adjustable parameter $\varphi$, (Eq. 14) to perform further second-stage modeling, which requires calibration to align with the experimental results:

$$T_{total} = t_{aspiration} + t_{acc} \tag{13}$$

$$E = E_{aspiration} + (\varphi - 1) \times E_{aspiration} \times H(t - t_{aspiration}) \tag{14}$$

where $t_{aspiration}$ is the time point that the cell gets aspirated into the micropipette, and $H$ is the Heaviside function. $t_{aspiration}$ in the simulation is 0.065 ms and $\varphi$ is assumed 11. The time for stiffening stage is $t_{acc}$. Stiffening was simulated for the RBC aspirated by $\theta = 10°$ micropipette (Eq. 13 and 14).

## Patch-clamp recording

HEK293T PIEZO1-OE cells were harvested and placed in a recording chamber containing 145 mM NaCl, 3 m M KCl, 1 mM $MgCl_2$ and 10 mM HEPES (pH 7.2, adjusted using NaOH) for patch-clamp analysis. Cells with a density of ~10,000 were seeded on coverslips for patch-clamp analysis on attached cells. HSPC was employed to apply negative pressure.

Micropipettes were fashioned similarly to those used in the fMPA assay, yielding an electrode resistance of 1.5-2.5 MΩ. EGTA was added to control the level of $Ca^{2+}$ inside the micropipette. Single-channel PIEZO1 currents were then amplified using the Axo-Patch 200B amplifier (Molecular Devices, US). All data were acquired at a sampling rate of 10 kHz with 1-2 kHz filtration, using the Digidata 1550 A Data Acquisition System (Molecular Devices, US). Analysis was conducted with pCLAMP10 software (Molecular Devices, US). Boltzmann plots were derived by fitting open probability $P_o \sim I/I_{max}$ versus negative pressure using the following function:

$$\frac{P_o}{1 - P_o} = e^{\alpha(\Delta p - P_{1/2})} \tag{15}$$

where $\Delta p$ is the negative pressure generated by the HSPC, $P_{1/2}$ is the pressure at which $P_o = 0.5$, and $\alpha$ is the slope of the plot reflecting the channel mechanosensitivity[25].

## Statistics and reproducibility

All measurements were taken from distinct samples. The representative results in the figures were from at least 3 independent experiments with similar results. All statistical analyzes were performed using Prism 9 (GraphPad Software, US). Welch's $t$-test (unpaired, two-tailed, not assuming equal standard deviation) was employed to evaluate the statistical significance. In instances where three or more groups were compared, Welch's ANOVA was implemented. Aspiration measurements were performed independently three to five repeats on separate days.

Pearson's test was performed to examine the result correlation between fMPA and FLIM assay (Fig. S5). The correlation coefficient, $r$ was calculated:

$$r = \frac{\sum(X_i - \bar{X})(Y_i - \bar{Y})}{\sqrt{\sum(X_i - \bar{X})(Y_i - \bar{Y})}} \tag{16}$$

where $X_i$ and $Y_i$ are the individual points for the two variables. $\bar{X}$ and $\bar{Y}$ are the mean of $X$ and $Y$, respectively.

## Reporting summary

Further information on research design is available in the Nature Portfolio Reporting Summary linked to this article.

## Data availability

The authors declare that the data supporting the findings of this study are available within the paper and its supplementary information files. The relevant raw data from each figure are provided in the Source data file. Source data are provided with this paper.

## Code availability

The FEA simulations were performed with commercial software ANSYS LS-DYNA R13.1.0. No special codes were developed for this study. The computation details are described in the Methods section.

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

## Acknowledgements

We thank Yingying Su and Neftali Flores Rodriguez for their imaging technical support of FLIM imaging and spinning disk confocal microscopy and acknowledge the technical and scientific assistance of Sydney Microscopy & Microanalysis, the University of Sydney node of Microscopy Australia, and Core Research Facility. We thank Boris Martinac and Jinyuan Vero Li for providing support on HEK293T cell lines and patch clamp recording. We thank Shaopeng Chi and Navid Bavi for providing technical support on patch clamp recording and data analysis. We thank Marc Ellis and HRI Thrombosis Group for animal support. We thank Ann Na Cho, Fengtao Jiang, and Yunduo (Charles) Zhao for providing critical suggestions; Zichen Zichri Li and Yiyao Catherine Chen for micropipette experimental discussion; and Yufei (Patrick) Bao and Yingqi (Kaitlyn) Zhang for maintaining the HEK293T cell lines. This work was supported by the Australian Research Council (ARC, DP200101970 – L.A.J. and Q.P.S.; DP240101768 – L.A.J. and C.D.C. and DP240102315 – L.A.J.); the Australian National Health and Medical Research Council (NHMRC, APP2003904 – L.A.J., C.D.C., and Q.P.S.); NSW Cardiovascular Capacity Building Program (Early-Mid Career Researcher Grant – L.A.J.); MRFF Cardiovascular Health Mission Grants (MRF2016165 – L.A.J.); Tour de Cure Annual Grant Pioneering Research (RSP-391-FY2023 – L.A.J.); Office of Global and Research Engagement (University of Sydney–University of Glasgow Ignition Grant – L.A.J.). National Heart Foundation Vanguard Grant (106979 – L.A.J.) L.A.J. is a Snow Medical Research Foundation Fellow (2022SF176) and a National Heart Foundation Future Leader Fellow Level 2 (105863). Q.P.S. is an NHMRC Emerging Leadership Fellow (APP1177374).

## Author contributions

H.J.W., Q.P.S., and L.A.J. designed the research and wrote and revised the manuscript. H.J.W., Y.W., T.A., L.M., B.R., Z.Z., and C.D.C performed research and analyzed data, conducted analysis and interpretation of data, and co-designed research; S.M. and P.V. performed the computational simulation; Q.L. provided critical guidance for the FEA simulations; Y.W. and J.J. provided technical support, maintained the cell line, and performed experiments. L.A.J. and Q.P.S. are the senior authors.

## Competing interests

The authors declare no competing interests.
