## [Peer Review File · Nature Communications]

Microscale geometrical modulation of PIEZO1 mediated mechanosensing through cytoskeletal redistributionREVIEWER COMMENTS

Reviewer #1 (Remarks to the Author):

This manuscript describes experiments on the gating of Piezo ion channels in pressurized membrane patches. In particular, data is presented that indicates that the micropipette diameter and opening angle, as well as cytoskeletal organization, can affect Piezo gating. The authors also used a continuum elastic model to make statements about how the membrane stress might distribute in the aspirated membrane patch. The most interesting aspect of the paper is, perhaps, the observed dependence of F-actin accumulation on the tip angle.

While the general issue of how constraints on the membrane affect the gating of Piezo ion channels is timely and interesting, the experiments presented here only offer very qualitative insights. Furthermore, the modeling presented here is both too detailed and too idealized to provide any real insight---it is too detailed in the sense that it involves many effectively free parameters that make it impossible to make any definite predictions, while it is too idealized in the sense that it does not seem to correctly capture the physics of membrane tension propagation and Piezo gating relevant for the specific experiments described here.

- In their modeling, the authors put much emphasis on the von Mises stress distribution in the membrane, which raises several issues: (1) The authors generally only show normalized stresses, from which it is not clear that the results even pass a basic sanity check---generically, lipid bilayers can only support stresses of a few percent. (2) There are two different kinds of membrane tension---the "mechanical" membrane tension the authors (implicitly) consider here and membrane tension connected to local membrane bending/unbending, which does not involve a change in membrane area. The latter kind of membrane tension tends to be energetically much less costly and is most relevant for membranes that show roughness due to, e.g., thermal fluctuations, heterogeneous lipid compositions, protein-induced membrane deformations, or membrane-cytoskeletal interactions. It is this latter kind of membrane tension that seems to underlie Piezo gating, with Piezo being curved in its closed state and Piezo unbending during gating. The mechanical membrane tension focused on by the authors does not seem to be the relevant membrane tension to consider here. (3) Given experimental and modeling uncertainties, it is unclear what the precise connection between the von Mises stresses calculated here and the (mechanical) membrane tension might be. (4) The distribution of membrane tension over the scale of microns considered by the authors will be strongly affected by membrane composition---in particular, the presence and organization of membrane proteins---as well as membrane-micropipette interactions. It is unclear why the stress distributions calculated by the authors should have any relevance to the membranes studied experimentally.

- It has been known for some time that Piezo gating is affected by force on the membrane/the channel and cytoskeletal attachments (e.g., ref. 26 in the manuscript and Nature Communications 7.1 (2016): 12939---a paper that should be discussed in this context). The present manuscript adds a twist to this, by putting emphasis on the micropipette shape. To gain new, solid understanding of Piezo gating in cell membranes, quantitative data on the molecular mechanisms underlying the observed behavior would be required. For instance, there are many ways in which the micropipette might affect membrane tension---e.g., through the physics of membrane-pipette adhesion, the constrained geometry at the tip of the micropipette, patch-clamp induced changes in membrane organization, etc. Furthermore, in addition to changes in the local membrane tension, Piezo gating is also affected by the local membrane geometry---see, e.g., eLife 7 (2018): e41968---and it is not clear whether the results in the present manuscript are due to changes in the local membrane tension (as suggested by the authors) or due to the local membrane geometry (or a combination of both).

- Especially in the discussion section, the manuscript makes strong claims that are not sufficiently substantiated by the data or the modeling, given their very qualitative nature, thus overselling the results of this study. Moreover, some of the language used here, starting with the term "buckle" in the title, is rather sloppy; "Buckling" has a well-defined meaning in science, and the paper does not provide much evidence that this is indeed what is going on here. Also, the paper starts off by talking about the binding site-driven or membrane curvature-driven Piezo localization described in refs. 19, 20 in the manuscript, while the manuscript argues (based on somewhat flimsy evidence) that the observed effects are driven by changes in the local membrane tension. This is confusing, as membrane geometry, Piezo binding, and local changes in membrane tension provide distinct physical mechanisms potentially affecting Piezo gating (see also the point above).

Reviewer #2 (Remarks to the Author):

Summary

In this manuscript, Wang et al develop combine multi-channel imaging and micropipette aspiration to study how local membrane deformation activates the mechanically activated ion channel Piezo1.

Combining this approach with computational analysis to calculate tension changes at the deformed membranes, the authors find that local Piezo1 responses scale with the pipette diameter and tip angle. Moreover, they discover a rapid actin redistribution into a buckle in the region of the cell right outside the pipette tip. This redistribution is in part Piezo1-dependent and promotes further channel activation. Importantly, contractility is dispensable for actin buckle amplification of Piezo1 function, pointing at actin accumulation as a mechanical input by itself.

The article is clearly written and illustrated, and its solid approach tackles an important issue regarding a channel family attracting a lot of attention in recent years: how are these channels activated by the cell environment? These findings seem especially relevant for RBC deformation by vessels and downstream volume regulation, known to depend heavily on Piezo1. However, the same could be said for white blood cells, and metastatic cancer cells both during blood-borne dispersion or invasion. In addition, one of the answers of this paper has relevant consequences for us Piezo researchers: we need to consider how we are polishing and holding our patch clamp pipettes in our daily experiments. These examples signify the broad relevance of this work

Overall, I think this paper should be published after addressing some comments.

Minor comments

- This is clearly out of the scope of this paper and should not affect any editorial decision. Just curiosity: do the authors have any experiment using GOF Piezo1 mutants known to control RBC volume and cause xerocytosis (Cahalan, eLife, 2015 and references therein)? Is their method capable of capturing these differences? Are the mutations shifting any of the curves: diameter, angle? What happens to actin in those cells?
- Actin buckle formation requires Piezo1. Describing the mechanism linking Piezo1 to Arp2/3-dependent F-actin (based on the CK666 experiments) seems out of the scope of the paper. However, the fact that Piezo1 triggers its actin-based own amplifier seems very relevant and should be discussed more in detail.
- Figure 1A shows nucleated cells and RBCs in the cross-section of the vessel (top right) but seems to focus on nucleated cells in the longitudinal cut (bottom left). However, most experiments use RBCs (it's one of the paper keywords!). Just showing RBCs would make things clearer.
- Line 240-241: If I got this right, which further substantiating should read which further substantiated or suppress which.
- Line 244: accumulates should read accumulate
- Line 254: accumulation formation sounds redundant to me. Accumulation should be enough.
- I think the title of Fig.S3 needs rewriting.

Dear Reviewers,

We express our gratitude for the thoughtful critiques and constructive comments, which have been pivotal in enhancing the manuscript's clarity and significance. For the past three months, we have undertaken comprehensive revisions, performed new experiments and computational modeling in response to the feedback, ensuring that each point raised has been meticulously addressed. **The major revision and changes are outlined as below:**

- **Aspirated cell membrane tension mapping through Fluorescence Lifetime Imaging Microscopy (FLIM):** We set up the concurrent FLIM imaging with our fluorescent micropipette aspiration assays (fMPA) and devised a new protocol using Flipper-TR[®] - Live cell membrane tension probe to map the tension distribution from the aspirated red blood cell's tongue to body (revised Fig. 3d-h and Fig. R3).
- **Elastic shell finite element analysis (FEA) to simulate red blood cell aspiration by micropipettes with various tip angles:** We applied new FEA analysis using elastic shell framework and rationale. The simulated maximum principal stress distribution in the aspirated tongue well correlated with the experimental membrane tension mapping by FLIM (revised Fig. 3a-c and Figs. R1 and 2)
- **Article Title Revised:** "Microscale geometrical modulation of PIEZO1 mediated mechanosensing through cytoskeletal redistribution"
- **Abstract Revision:** "...increased micropipette tip angles intensify PIEZO1 activation, linked to a significant reorganization of F-actin, resulting in physical constrictions that amplify the channel's activity in regions undergoing tension stress..."
- **Introduction Update:** "The inherent curvature of PIEZO1 trimers conveys sensitivity to tension, further influenced by their distinct architecture and size. PIEZO1's role in mechanosensing is highlighted by its preferential localization within the biconcave dimples of red blood cells, crucial for erythrocyte function."
- **Figures Adjusted:**
 - Figure 1 – Updated to include a red blood cell, aligning with reviewer feedback.
 - Figure 3 – Enhanced with new simulation data from elastic shell FEA study and quantitative membrane tension analysis.
 - Figure 5 – Title modified to "F-actin accumulation influences PIEZO1 activity within the cellular protrusion."
 - Figure S3 – Caption revised to "Calcium mobilization in *Piezo1*-KO^{RBC} under aspiration."
- **Main Text Revisions:**
 - Substituted "buckle" terminology with "constriction" to avoid misinterpretation.
 - Added a new paragraph in Section 2.3 detailing membrane tension findings and their implications on PIEZO1 activation.
 - Grammatical corrections implemented throughout the manuscript.
- **Conclusion and Discussion Enhancement:** "Our combined FLIM tension mapping and elastic shell FEA simulations demonstrate that increased membrane tension, particularly around the aspirated cell's neck region, aligns with the observed upsurge in PIEZO1-mediated calcium influx. This supports the hypothesis that microscale geometrical determinant, i.e. micropipette tip angle and subsequent physical constrictions are critical in modulating mechanosensitive ion channel PIEZO1's mechano-gating."

Point-by-point responses to the reviewers' comments

Our point-by-point responses are denoted in blue, while the corresponding actions are detailed in purple. Changes within the revised manuscript are marked in red for easy identification.

Reviewer #1 (Remarks to the Author):

This manuscript describes experiments on the gating of Piezo ion channels in pressurized membrane patches. In particular, data is presented that indicates that the micropipette diameter and opening angle, as well as cytoskeletal organization, can affect Piezo gating. The authors also used a continuum elastic model to make statements about how the membrane stress might distribute in the aspirated membrane patch. The most interesting aspect of the paper is, perhaps, the observed dependence of F-actin accumulation on the tip angle.

Response: We extend our gratitude for the affirmative remarks and constructive comments regarding our exploration of the micropipette geometry, F-actin redistribution, and PIEZO1 ion channel activation dynamics.

While the general issue of how constraints on the membrane affect the gating of Piezo ion channels is timely and interesting, the experiments presented here only offer very qualitative insights. Furthermore, the modeling presented here is both too detailed and too idealized to provide any real insight---it is too detailed in the sense that it involves many effectively free parameters that make it impossible to make any definite predictions, while it is too idealized in the sense that it does not seem to correctly capture the physics of membrane tension propagation and Piezo gating relevant for the specific experiments described here.

Response: Upon reflection, we acknowledge the concerns regarding the qualitative nature of our findings and the potentially abstract nature of our modeling. In response, we have:

1. Refined our computational model using an updated inverse finite element analysis (FEA) technique to determine the material parameters for creating an elastic shell model. This has allowed us to more accurately simulate the biomechanical behavior of RBCs under differential micropipette aspiration angles ($\theta = 0^\circ, 5^\circ, \text{ and } 10^\circ$), providing a more realistic representation of the tension dynamics post-aspiration. As a result, the updated model correlates with experimental observations of RBC deformation closely, enhancing the predictive power of our simulations.

2. Augmented our methodology with a comprehensive Fluorescence Lifetime Imaging Microscopy (FLIM) study. Employing a membrane tension-sensitive probe, Flipper-TR® (Colom, Derivery et al. 2018), we have quantified the distribution of membrane tension across RBCs subject to micropipette aspiration. This has more precisely elucidated the relationship between micropipette tip angle and PIEZO1-mediated calcium mobilization, affirming our hypothesis that membrane tension is a critical factor in physical constriction induced PIEZO1 activities.

- In their modeling, the authors put much emphasis on the von Mises stress distribution in the membrane, which raises several issues: (1) The authors generally only show normalized stresses, from which it is not clear that the results even pass a basic sanity check---generically, lipid bilayers can only support stresses of a few percent.

Response: We appreciate the reviewer's critical insights on the presentation of von Mises stress distributions in our preliminary modeling work. To clarify, we have refined our FEA approach in the following ways:

1. We have calibrated our cell aspiration modeling parameters against the real experiments to ensure the physiological relevance (Fig. R1). Specifically, we have imposed constraints on the stress values to reflect the limited stress capacity of lipid bilayers, addressing the sanity check concerns.

Figure R1 (revised Figure S13). Modelling red blood cell deformation upon micropipette aspiration. **a** Initial setup for the RBC aspiration modeling, displaying the undeformed biconcave geometry of a real RBC as used in the computational simulations. The undeformed state serves as a baseline for subsequent deformation analysis. **b** Snapshots of RBC aspirated

by $\theta = 0^\circ$. Measurement was performed using ImageJ after cell mask segmentation. **c** Elastic shell finite element analysis (FEA) modeling schematic. Biconcave shaped RBC was placed next to the micropipette first, representing the initial stage. Then aspirated RBC dimension was reconstructed and meshed.

2. Our revised FEA method now includes an elastic shell model, which better accounts for the complex biomechanical behaviors of RBCs under various aspiration angles. This improvement provides a more realistic representation of the stress distribution in the cell membrane (Fig. R2).

Figure R2 (revised Figure 3f-h). **a-c** Front view of the FEA simulated aspirated RBC maximum principal stress contours, denoted the change of membrane tension distribution on the aspirated RBCs. Values are normalized based on the maximum principal stress value of all conditions. Black arrow pointed to the regions with the highest membrane tension when the cell was aspirated by $\theta = 0^\circ$ (**a**) 5° (**b**) and 10° (**c**).

Action taken:

1. The Method section in the revised manuscript has been updated with the new section ‘Elastic shell finite element analysis on single cell micropipette aspiration’:

Page 29 line 13 – Page 31 line 12: *‘The simulation of micropipette aspiration was conducted using LS-DYNA (ANSYS, US), a commercially available finite element method (FEA) analysis software. RBCs were considered as an elastic shell with thickness of 200 nm to ensure deformability during simulation (Fig. S13a). The material was considered isotropic and thus, the stress-strain relationship was expressed as:*

$$\sigma = \mathbf{C} : \varepsilon \quad (5)$$

where σ and ε are the three dimensional stress and strain tensor, respectively (Bower 2009). The stress can be written as:

$$\boldsymbol{\sigma} = \begin{bmatrix} \sigma_{xx} & \sigma_{xy} & \sigma_{xz} \\ \sigma_{xy} & \sigma_{yy} & \sigma_{yz} \\ \sigma_{xz} & \sigma_{zy} & \sigma_{zz} \end{bmatrix} \quad (6)$$

The fourth-order elasticity tensor \mathbf{C} can be simplified as:

$$\mathbf{C} = \frac{E}{(1+\nu)(1-2\nu)} \begin{bmatrix} 1-\nu & \nu & \nu & 0 & 0 & 0 \\ \nu & 1-\nu & \nu & 0 & 0 & 0 \\ \nu & \nu & 1-\nu & 0 & 0 & 0 \\ 0 & 0 & 0 & \frac{1-2\nu}{2} & 0 & 0 \\ 0 & 0 & 0 & 0 & \frac{1-2\nu}{2} & 0 \\ 0 & 0 & 0 & 0 & 0 & \frac{1-2\nu}{2} \end{bmatrix} \quad (7)$$

where E is the Young's modulus (120 Pa), ν is the Poisson's ratio (0.49). During the initial stage of simulation, RBC geometry was considered based on a continuous degree-4 surface which is constrained by three principal dimensions: the main diameter d , the thickness at the dimple centre, b , and the maximum thickness, h (Kuchel, Cox et al. 2021). The geometry is reconstructed based on the equation:

$$(x^2 + y^2 + z^2)^2 + P(x^2 + y^2) + Qz^2 + R = 0 \quad (8)$$

where,

$$P = -\frac{d^2}{2} + \frac{h^2}{2} \left(\frac{d^2}{b^2} - 1 \right) - \frac{h^2}{2} \left(\frac{d^2}{b^2} - 1 \right) \left(1 - \frac{b^2}{h^2} \right)^{\frac{1}{2}} \quad (8.1)$$

$$Q = P \frac{d^2}{b^2} + \frac{b^2}{4} \left(\frac{d^4}{b^4} - 1 \right) \quad (8.2)$$

$$R = -P \frac{d^2}{4} - \frac{d^4}{16} \quad (8.3)$$

The geometrical constraints are considered to be $d = 8$, $b = 1$, and $h = 2.12 \mu\text{m}$ (Fig. S12a).

Before performing stress analysis, we utilized the inverse FEA to obtain the material properties of the human RBC. Thereby, we measured the deformation of the RBC aspirated by the $\theta = 0^\circ$ (Fig. S13b) and then evaluated the material properties of the aspirated cell inversely. The force applied to the RBC was considered as $\Delta p = -25 \text{ mmHg}$ which was corresponded to the experiment results, while the aspirated tongue length in the simulation correlated with the experiment measurement to derive the material properties (E and ν). After obtaining the material properties, we confirmed that the tongue protrusion length was well aligned with the

experiment measurements (Fig. S13c) and thereby, the extracted material parameters and protrusion length were then implemented to the simulation at tip angle $\theta = 0^\circ, 5^\circ, \text{ and } 10^\circ$ as boundary condition. Of note, an aspirating pressure $\Delta p = -25 \text{ mmHg}$ was always applied to the nodes at the aspirated dome during the explicit dynamic analysis.

Since the micropipette is significantly stiffer than red blood cell, the glass micropipette was considered as a rigid body. A fillet radius was applied to the tip of the micropipette to mimic the experimental setups and avoid high element distortion and contact singularity. The frictionless boundary condition was applied to the cell and the adjacent micropipette wall since BSA was included during aspiration experiments. Pure penalty boundary was applied to the contact between the micropipette and the cell. while the body was fixed when mechanical anchor was applied. In order to compare the results, maximum principal stress was calculated for each studied case (Bavi, Nakayama et al. 2014) that can be calculated from

$$\det(\boldsymbol{\sigma} - \lambda \mathbf{I}) = 0 \quad (9).$$

The rest are the shear stress during deformation. The values reflected the trend of membrane tension elevation due to the aspiration (Bavi, Nakayama et al. 2014) and were normalized based on the maximum value of all conditions at each cell type.”

Meanwhile, to avoid the confusion raised by the wording, we redefine the stress change we found here as the “normalized change of membrane tension from the resting state”. The details are below:

2. In the revised figures (Figs. 3a-c), we present the recalibrated stress distribution with explicit mention of the physiological limits to which lipid bilayers are subjected, thus ensuring the results are within a credible range.
3. The manuscript text has been updated to explicitly state that the maximum principal stress values are normalized to the maximum stress supported by lipid bilayers, providing a clear reference frame for the reported data.

(2) There are two different kinds of membrane tension---the "mechanical" membrane tension the authors (implicitly) consider here and membrane tension connected to local membrane bending/unbending, which does not involve a change in membrane area. The latter kind of membrane tension tends to be energetically much less costly and is most relevant for membranes that show roughness due to, e.g., thermal fluctuations, heterogeneous lipid compositions, protein-induced membrane deformations, or membrane-cytoskeletal interactions. It is this latter kind of membrane tension that seems to underlie Piezo gating, with Piezo being curved in its closed state and Piezo unbending during gating. The mechanical membrane tension focused on by the authors does not seem to be the relevant membrane tension to consider here.

Response: We thank the reviewer for the insightful comments. We agree that local membrane bending/unbending is one of the primary factors that drive the PIEZO1 gating. However, studying the membrane roughness and delving into how local membrane shape changes, particularly curvature changes at the scale of <50 nm (Ridone, Grage et al. 2018), is not the focus of this study. Here, we used the ‘force-from-lipid’ model to explain how the membrane tension mediates PIEZO1 ion channel activation, thus modulating calcium mobilization in the aspirated cells. Importantly, this ‘force-from-lipid’ model has been considered by multiple research groups in this field as one of the well-recognized models to elucidate the PIEZO1 activity under mechanical force stimulation (Young, Lewis et al. 2022, Bavi, Cox et al. 2023). Moreover, we have acknowledged that the local membrane tension (i.e., membrane footprint and local curvature) also mediate the PIEZO1 activity (Haselwandter and MacKinnon 2018). However, it's essential to emphasize that the local membrane bending and macroscopic cell deformation (representing two distinct types of induced tension) do not operate independently; rather, they are closely intertwined on numerous levels (Rangamani, Mandadap et al. 2014, Shi and Baumgart 2015, Golani, Ariotti et al. 2019). Again, given that the aim of this study is to explore the impact of ‘physical constriction’ on PIEZO1 gating and activity, we will focus on the macroscopic deformation of the cells under micropipette aspiration , a scenario where ‘force-from-lipid’ model is predominantly applied.

Action taken:

To make it clear, we have incorporated a section to discuss the potential contributions of membrane tension, local curvature, composition, and bonded proteins to the PIEZO1-mediated calcium mobilization observed in this study:

Page 20 Line 9-33 *“Our combined FLIM tension mapping and elastic shell FEA simulations demonstrate that increased membrane tension, particularly around the aspirated cell’s neck region, aligns with the observed upsurge in PIEZO1-mediated calcium influx. This supports the hypothesis that microscale geometrical determinant, i.e. micropipette tip angle and subsequent physical constrictions are critical in modulating mechanosensitive ion channel PIEZO1’s mechano-gating.”* *“In the meantime, the induction of membrane curvature by PIEZO1 provides a potential physical determinant for its subcellular distribution. PIEZO1 has been observed to display a polarized distribution in migrating keratinocytes to be enriched at focal adhesion sites(Yao, Tijore et al. 2022), and to preferentially locate within the biconcave dimples of RBCs(Vaisey, Banerjee et al. 2022), emphasizing its significant role in RBC mechanosensing, a crucial aspect of erythrocyte function. Studies have reported that PIEZO1 activities are not only mediated by the local membrane tension(Bavi, Cox et al. 2023), but also influenced by the local membrane footprint, or membrane curvature(Haselwandter and MacKinnon 2018, Young, Lewis et al. 2022). More importantly, the resting membrane tension before any mechanical stimulation(Lewis and Grandl 2015), the cluster and density of the channels(Lewis and Grandl 2021) can also alters the sensitivity and activities of the channels. Therefore, it is also important to take these factors into consideration in the further study when untacking the microgeometry effect on single cell mechanosensing.”*

(3) Given experimental and modeling uncertainties, it is unclear what the precise connection between the von Mises stresses calculated here and the (mechanical) membrane tension might be. (4) The distribution of membrane tension over the scale of microns considered by the authors will be strongly affected by membrane composition---in particular, the presence and organization of membrane proteins---as well as membrane-micropipette interactions. It is unclear why the stress distributions calculated by the authors should have any relevance to the membranes studied experimentally.

Response: Although local membrane geometry modulation is not investigated in this study, we have attempted to quantify the change of local membrane tension in the aspirated cell by performing the Fluorescence Lifetime Imaging Microscopy (FLIM) assay. In this assay, we used the mechanical tension-dependent Flipper-TR probe (Colom, Derivery et al. 2018), which responds to the push-and-pull effect in the lipid bilayer when the membrane experiences tension, and planarizes to exhibit an increased fluorescence lifetime (τ , **Fig. R3**) when the membrane tension is elevated. The FLIM images demonstrated that an increased tip angle θ

promotes a rapid membrane tension escalation, specifically in the aspirated cell's tongue region (Figs. R1 a-c). As a result, higher PIEZO1 activities and Ca^{2+} mobilization was observed (proven by the fMPA in the main manuscript). Thus, aspirating cell membrane can escalate local membrane tension, indicated by an increasing fluorescence lifetime (Fig. R3d). We have added the following figure in the main manuscript:

Figure R3 (revised Figure 3 a-e) Membrane tension mapping of RBC aspirated by micropipettes with different tip angles. **a-c**, top panel: Representative fluorescence lifetime mapping of individual RBCs labelled with Flipper-TR and aspirated at $\Delta p = -25$ mmHg. The RBCs were aspirated by micropipettes with $\theta = 0^\circ$ (**a**), 5° (**b**) and 10° (**c**), respectively. bottom panel: Fluorescence lifetime distribution of aspirated cell body (orange) and tongue (red) regions were plotted and fitted with Gaussian distribution. **d**, statistics of Flipper-TR lifetime of RBCs aspirated by micropipettes with different angles. The lifetime of the cell body and tongue regions were quantified separately. A major increase in the lifetime was noticed in both body and tongue regions when the tip angle θ increased. **e** the ratio of tongue-to-body lifetime

on RBC aspirated by micropipettes with different tip angles. At least 35 cells were measured to obtain mean \pm s.e.m. assessed by unpaired, two-tailed Student's t-test.

It is noteworthy that when the tip angle increases, the Flipper-TR lifetime increases in both body (**Fig. R3d**; $\tau = 4.7 \pm 0.1$ ns at $\theta = 0^\circ$, $\tau = 5.4 \pm 0.1$ ns at $\theta = 5^\circ$, and $\tau = 5.4 \pm 0.1$ ns at $\theta = 10^\circ$) and tongue regions ($\tau = 4.8 \pm 0.1$ ns at $\theta = 0^\circ$, $\tau = 5.6 \pm 0.2$ ns at $\theta = 5^\circ$, and $\tau = 5.9 \pm 0.1$ ns at $\theta = 10^\circ$). Since there is a linear correlation between the fluorescence lifetime and the membrane tension (Colom, Derivery et al. 2018), our results provide clear evidence that increasing the tip angle can rapidly increase the membrane tension. Intriguingly, the FLIM results also demonstrate that the membrane tension in the body is slightly higher than the tongue when the cell is aspirated by micropipette with $\theta = 0^\circ$ (**Fig. R3e**; ratio = 0.98 ± 0.01). This tension distribution shifted when the tip angle increased (ratio = 1.08 ± 0.01 at $\theta = 5^\circ$), and eventually, the tongue demonstrates a much higher tension when $\theta = 10^\circ$ (ratio = 1.14 ± 0.01). These FLIM measurements reveal a well-aligned trend with our fMPA results (**Fig. R4a**; $r = 0.83$, p -value = 0.0429) and previous mentioned simulation in the aspirated tongue (**c.f. Fig. R2a-c & Fig. R3. a-c; Fig. 4b**; $r = 0.99$). Since it has been reported that increasing the membrane tension by ~ 3 -fold would facilitate PIEZO1 from limited gating to fully gating (Cox, Bae et al. 2016), the change of tension detected by the FLIM assay here further demonstrates the statement we made in the original manuscript – *“increasing tip angle would be the major cause of the elevated calcium influx during aspiration, especially in the tongue region.”*

Moreover, our revised elastic shell FEA modeling now incorporates a refined calculation of maximum principal stress (Bavi, Nakayama et al. 2014) to more accurately represent the mechanical stress distribution within the membrane, considering the macroscopic forces applied during micropipette aspiration. This revised approach enhances the relevance of our model to the physical conditions experienced by the cell.

Notably, our simulation results explained a much higher membrane tension will be generated at the aspirated cell tongue (Figs. 3a-c), which is further supported by the FLIM tension measurement. Since elevating membrane tension is well-known as one of the major causes to the PIEZO1 gating under mechanical stimulation (Cox, Bavi et al. 2019, Young, Lewis et al. 2022), our finding imposed that increasing tip angle would be the major cause of the elevated calcium influx during aspiration, especially in the tongue.

Figure R4 (revised Figure S5) Correlation Between PIEZO1 Activation versus Membrane Tension, and Simulated Stress versus Membrane Tension. Pearson correlation tests was performed between FLIM versus fMPA (a) and FEA versus FLIM (b) results at $\Delta p = -25\text{mmHg}$. Statistical outcome demonstrated a strong positive correlation between FLIM versus fMPA results ($r = 0.83$) and FEA versus FLIM results ($r = 0.99$). The correlation underscores our hypothesis that tip angle variation is a key mechanical factor influencing PIEZO1 channel activity, further supported by FLIM data and FEA results demonstrating the tension increase with larger tip angles.

Action taken: The new FLIM data has been added and described in the Result section 2.3 with updated Fig.3 in the revised manuscript.

Page 9 Line 14 – Page 10 Line 3 “To discern the effect of microscale geometrical variations on PIEZO1 channel activity, we employed the fluorescence lifetime imaging microscopy (FLIM) to examine the membrane tension of the RBCs by using tension-sensitive probe, Flipper-TR[®] (Figs. 3a-c). This tension imaging probe responds to the push-and-pull effect in the lipid bilayer when the membrane experiences tension and planarize to exhibit an increased fluorescence lifetime τ when the membrane tension is elevated. As the result, it is clear that aspirating cells can increase the membrane tension, indicated by an increasing fluorescence lifetime (Fig. 3d). More importantly, It is noteworthy that when the tip angle increases, the Flipper-TR[®] lifetime increases in both body (Fig. 3d; $\tau = 4.7 \pm 0.1$ ns at $\theta = 0^\circ$, $\tau = 5.4 \pm 0.1$ ns at $\theta = 5^\circ$ and $\tau = 5.4 \pm 0.1$ ns at $\theta = 10^\circ$) and tongue ($\tau = 4.8 \pm 0.1$ ns at $\theta = 0^\circ$, $\tau = 5.6 \pm 0.2$ ns at $\theta = 5^\circ$ and $\tau = 5.9 \pm 0.1$ ns at $\theta = 10^\circ$) regions. Since there is a linear correlation between the fluorescence lifetime and the membrane tension³⁴, our results provide clear evidence that increasing the tip angle would rapidly increase the membrane tension. After performed the Pearson’s test, the FLIM data shows a strong positive correlation with the fMPA results (Fig.

S5a; $r = 0.83$, p -value = 0.0429), providing strong evidence that increasing micropipette tip angle would increase the membrane tension in the aspirated cell, especially in the aspirated tongue, to promote the PIEZO1 mediated calcium influx.

Intriguingly, the FLIM results also demonstrate that the membrane tension in the body is slightly higher than the tongue when the cell is aspirated by a $\theta = 0^\circ$ micropipette (Fig. 3e; ratio = 0.98 ± 0.01). This tension distribution shifted when the tip angle increased (ratio = 1.08 ± 0.01 at $\theta = 5^\circ$), and eventually, the tongue experienced a higher tension when $\theta = 10^\circ$ (ratio = 1.14 ± 0.01).”

- It has been known for some time that Piezo gating is affected by force on the membrane/the channel and cytoskeletal attachments (e.g., ref. 26 in the manuscript and Nature Communications 7.1 (2016): 12939---a paper that should be discussed in this context). The present manuscript adds a twist to this, by putting emphasis on the micropipette shape. To gain new, solid understanding of Piezo gating in cell membranes, quantitative data on the molecular mechanisms underlying the observed behavior would be required. For instance, there are many ways in which the micropipette might affect membrane tension---e.g., through the physics of membrane-pipette adhesion, the constrained geometry at the tip of the micropipette, patch-clamp induced changes in membrane organization, etc.

Response: Thank you for highlighting the intricate factors influencing Piezo gating. Our study indeed brings attention to the micropipette shape, adding a novel perspective to the mechanical environment's role in channel gating. We acknowledge the complexity of micropipette-membrane interactions and their potential to affect membrane tension through various mechanisms. While our current data does not dissect these mechanisms in detail, it provides a foundational understanding of the geometric influence on PIEZO1 activation.

Delving into the detailed gating mechanism of PIEZO1 on the magnitude of <50 nm curvature change is out of the scope of this study (Ridone, Grage et al. 2018). Furthermore, fMPA in this study differs from the traditional patch clamp, in which membrane-pipette adhesion is preferred to form a giga seal for the picoamp current measurement (Suchyna, Markin et al. 2009). In contrast, bovine serum albumin (BSA) is used in our assay to minimize the non-specific interaction between the aspirated cell and the glass (Fillafer, Mussel et al. 2018). Thereby, the adhesion energy between the membrane and the glass was minimized, and only

the physical constriction induced by the narrow geometry and the mechanical force applied by the aspiration would stimulate the cell.

In addition, we established a pulling protocol for a micropipette with new geometry to further explore how geometrical constriction from the narrow space would mediate the cellular mechano-response. Specifically, we post-processed the pulled micropipette with fire polish. First, we fabricated a parallel micropipette with an inner diameter of 3.5 μm . Then, the cut micropipette was mounted to the MicroForge to slowly melt the glass wall at the tip to decrease the inner diameter. After the inner diameter decreased to 1 μm , the heat was removed to cool and solidify the shape. As a representative outcome, a micropipette with an inner diameter $d = 1.12 \mu\text{m}$ at the tip was achieved (**Fig. R5a**), which exhibited a similar geometry at the tip compared to the normal micropipette with tip angle $\theta = 0^\circ$ (**Fig. 1c** in the revised manuscript). In sharp contrast, the inner diameter rapidly increased to $d = 3.47 \mu\text{m}$ after the tip region.

When this new holding micropipette geometry aspirated the RBC, the calcium intensity change increased rapidly, particularly in the body region (**Fig. R5b**). Remarkably, the intensity change is higher in holding micropipette aspirated RBC (**Fig. R5c**; 7.17 ± 1.59 fold at $\Delta p = -25 \text{ mmHg}$) than in normal $\theta = 0^\circ$ micropipette. Combining with our findings in the revised manuscript, we suggest that changing the later region diameter would upregulate the tension in the aspirated RBC, and therefore promote the PIEZO1-mediated calcium mobilization. However, such micropipette geometry is out of the scope of this study. Thus, we will not discuss it further in the revised manuscript.

Figure R5. RBC calcium mobilization when aspirated by a holding micropipette with modified geometry. **a** new pulling protocol helps fabricate a holding micropipette with a narrow opening. After cutting the micropipette to obtain an inner diameter of 3.5 μm , the micropipette was placed on a microforge to perform fire polishing. The micropipette was placed next to the heating source and the glass wall was melted to narrow down the tip slowly. The heat was stopped once the tip diameter decreased to 1 μm . As shown in the representative DIC image, the tip diameter is 1.12 μm . **b** Representative fluorescence snapshot of holding micropipette-aspirated RBC. **c**, Ca²⁺ intensity change at different aspiration pressure when the RBC is aspirated by holding a micropipette. As the pressure increases, the intensity change in the RBC aspirated by the holding pipette was larger than the RBC aspirated by a normal parallel micropipette with tip diameter $d = 1 \mu\text{m}$. Intensity fold change was measured from $n \geq 6$ aspirated RBCs to obtain mean \pm s.e.m.

On the other hand, further investigating the gating mechanism of PIEZO1 at the molecular scale is not the focus of this study. Instead, we performed studies to investigate how the structure of F-actin would alter its accumulation at the micropipette tip (**Fig. 5d** in the revised manuscript), in terms of actin length (Latrunculin A, inhibiting actin monomer binding into F-actin filament), F-actin branching (CK-666, inhibiting Arp2/3), and F-actin structure reinforcement (jasplakinolide, inducing actin polymerization). Importantly, our findings did not support that F-actin would directly engage with the PIEZO1 to modify the mechanical force transduction towards the channel, but rather proposed that accumulation would alter the

mechanical transduction across the whole cell membrane, further enhance the mechanical tension and promote PIEZO1 activities explained by the ‘force-from-lipid’ model (Bavi, Nakayama et al. 2014).

Furthermore, in addition to changes in the local membrane tension, Piezo gating is also affected by the local membrane geometry---see, e.g., eLife 7 (2018): e41968---and it is not clear whether the results in the present manuscript are due to changes in the local membrane tension (as suggested by the authors) or due to the local membrane geometry (or a combination of both).

Response: We are thankful for the reviewer providing this inspiring information. Despite the eLife 7 (2018): e41968, other studies have also reported that local membrane geometries (Haselwandter and MacKinnon 2018), channel density and cluster (Lewis and Grandl 2021), and resting membrane tension (Lewis and Grandl 2015) may also tune the channel gating. However, since our focus is not delving into the gating mechanism of the PIEZO1 channels, local membrane geometry at the nanometer scale and its effect on PIEZO1 gating will not be further explored in this study.

Action taken: Nevertheless, we have cited these works and included the discussion of this nanoscale geometry effect on PIEZO1-mediated mechano-response in the Discussion of the revised manuscript as stated in the previous response.

Page 20 Line 27 – 33 *“Studies have reported that PIEZO1 activities are not only mediated by the local membrane tension(Bavi, Cox et al. 2023), but also influenced by the local membrane footprint, or membrane curvature(Haselwandter and MacKinnon 2018, Young, Lewis et al. 2022) More importantly, the resting membrane tension before any mechanical stimulation(Lewis and Grandl 2015), the cluster and density of the channels(Lewis and Grandl 2021) can also alters the sensitivity and activities of the channels. Therefore, it is also important to take these factors into consideration in the further study when untacking the microgeometry effect on single cell mechanosensing.”*

- Especially in the discussion section, the manuscript makes strong claims that are not sufficiently substantiated by the data or the modeling, given their very qualitative nature, thus overselling the results of this study. Moreover, some of the language used here, starting with the term "buckle" in the title, is rather sloppy; "Buckling" has a well-defined meaning in science, and the paper does not provide much evidence that this is indeed what is going on here.

Response: We thank the reviewer for pointing out the misleading wording in the manuscript.

Action take: We replaced all "buckle" with "F-actin accumulation" or "physical constriction".

Also, the paper starts off by talking about the binding site-driven or membrane curvature-driven Piezo localization described in refs. 19, 20 in the manuscript, while the manuscript argues (based on somewhat flimsy evidence) that the observed effects are driven by changes in the local membrane tension. This is confusing, as membrane geometry, Piezo binding, and local changes in membrane tension provide distinct physical mechanisms potentially affecting Piezo gating (see also the point above).

Response: We agreed that talking about the impact of curvature on PIEZO1 in the Introduction may give readers misleading information.

Action taken: We moved the section to the Discussion.

Reviewer #2 (Remarks to the Author):

Summary: In this manuscript, Wang et al develop combine multi-channel imaging and micropipette aspiration to study how local membrane deformation activates the mechanically activated ion channel Piezo1.

Combining this approach with computational analysis to calculate tension changes at the deformed membranes, the authors find that local Piezo1 responses scale with the pipette diameter and tip angle. Moreover, they discover a rapid actin redistribution into a buckle in the region of the cell right outside the pipette tip. This redistribution is in part Piezo1-dependent and promotes further channel activation. Importantly, contractility is dispensable for actin buckle amplification of Piezo1 function, pointing at actin accumulation as a mechanical input by itself.

The article is clearly written and illustrated, and its solid approach tackles an important issue regarding a channel family attracting a lot of attention in recent years: how are these channels activated by the cell environment? These findings seem especially relevant for RBC deformation by vessels and downstream volume regulation, known to depend heavily on Piezo1. However, the same could be said for white blood cells, and metastatic cancer cells both during blood-borne dispersion or invasion. In addition, one of the answers of this paper has relevant consequences for us Piezo researchers: we need to consider how we are polishing and holding our patch clamp pipettes in our daily experiments. These examples signify the broad relevance of this work

Overall, I think this paper should be published after addressing some comments.

Response: We are grateful for Reviewer #2's positive comments. Please check our point-by-point response below.

Minor comments

- This is clearly out of the scope of this paper and should not affect any editorial decision. Just curiosity: do the authors have any experiment using GOF Piezo1 mutants known to control RBC volume and cause xerocytosis (Cahalan, eLife, 2015 and references therein)? Is their method capable of capturing these differences? Are the mutations shifting any of the curves: diameter, angle? What happens to actin in those cells?

Response: We thank the reviewer for such insightful questions and the opportunity to clarify our experimental approach regarding GOF PIEZO1 mutants. While our study does not include experiments with these mutants, we understand their potential impact on RBC volume regulation and membrane tension. Instead, we have used the PIEZO1 agonist YODA1 as a positive control to validate that all the calcium mobilization captured in this study is PIEZO1-dependant (**Fig. S2a** in the revised manuscript). YODA1 treatment, on the other hand, can be interpreted as an approach to mimic the performance of PIEZO1 GOF phenotypes in terms of gating since the agonist is known to lower the gating mechanical threshold for the channel (Syeda, Xu et al. 2015, Wang, Chi et al. 2018, Botello-Smith, Jiang et al. 2019). Here, we put the tip angle $\theta = 0^\circ$ as an example to show the comparison. It is clear that after $0.5 \mu\text{M}$ Yoda1 treatment, the Ca^{2+} intensity changes were upregulated, particularly in the pressure regime $\Delta p = -15$ to -30 mmHg (**Fig. R6**).

Figure R6. Yoda1-dependent Ca^{2+} mobilization on aspirated RBC change with different aspiration pressures. Intensity fold change was measured from $n \geq 6$ aspirated RBCs to obtain mean \pm s.e.m..

- Actin buckle formation requires Piezo1. Describing the mechanism linking Piezo1 to Arp2/3-dependent F-actin (based on the CK666 experiments) seems out of the scope of the paper. However, the fact that Piezo1 triggers its actin-based own amplifier seems very relevant and should be discussed more in detail.

Response: We thank the reviewer for this constructive suggestion. In fact, the finding we stated in this study cannot provide solid evidence that this is a linking mechanism between PIEZO1 and F-actin. In fact, we believe that F-actin accumulation at the neck region (i.e., part of the

aspirated cell at the micropipette tip) serves as a physical constriction to the membrane environment. Such constriction would change the membrane tension at the tongue to the dome region of the aspirated cell, and thus, mediate PIEZO1 ion channel activities by following the ‘force-from-lipid’ model. To this end, we utilized the CK666 and other F-actin targeting treatments to modify the structure and thereby modify the impact of the physical constriction to prove that the strength of the F-actin constriction is correlated with the amplification of PIEZO1 activities at the tongue region.

Regarding the driving factor of F-actin accumulation, Luo et al. (Luo, Mohan et al. 2013) has reported that the shear force inside the cell under aspiration would accumulate the actin filaments at the neck region. Our finding of increasing aspiration pressure promoted the chance of observing F-actin accumulation at the neck further supports the reported finding (**Figs. 4d-f** in the revised manuscript). Intriguingly, higher PIEZO1 expression level also favored the F-actin accumulation in our results but delving into details to prove that such upregulation is solely depended on the PIEZO1 channels and their activities, or upregulation is also mediated by the expression level-induced mechanical property change in the cell line is out of the scope of this study.

- Figure 1A shows nucleated cells and RBCs in the cross-section of the vessel (top right) but seems to focus on nucleated cells in the longitudinal cut (bottom left). However, most experiments use RBCs (it’s one of the paper keywords!). Just showing RBCs would make things clearer.

Response: We appreciate the attention to detail in our figures. We agree that consistency in the cellular focus of our images is important for clarity. To avoid confusion and maintain focus on the primary cell type used in our experiments, we have revised Figure 1A to exclusively show RBCs in both the cross-section and longitudinal views of the vessel.

Action taken: The nucleated cell in **Fig. 1a** has been replaced with RBC.

- Line 240-241: If I got this right, which further substantiating should read which further substantiated or suppress which.

Action taken: Page 12 Line 10 “substantiating” has been replaced with “substantiated”

- Line 244: accumulates should read accumulate

Action taken: Page 14 Line 7 “accumulates” has been replaced with “accumulate”

- Line 254: accumulation formation sounds redundant to me. Accumulation should be enough.

Action taken: Page 14 Line 17 and Line 19 “accumulation formation” has been replaced with “accumulation”

- I think the title of Fig.S3 needs rewriting.

Action taken: The title of Fig.S3 has been rewritten for accuracy and precision, now titled “*Piezo1-KO^{RBC} suppressed calcium mobilization in aspirated RBCs*”

Reference

- Bavi, N., C. D. Cox, Y. A. Nikolaev and B. Martinac (2023). "Molecular insights into the force-from-lipids gating of mechanosensitive channels." *Current Opinion in Physiology*: 100706.
- Bavi, N., Y. Nakayama, O. Bavi, C. D. Cox, Q. H. Qin and B. Martinac (2014). "Biophysical implications of lipid bilayer rheometry for mechanosensitive channels." *Proc Natl Acad Sci U S A* **111**(38): 13864-13869.
- Botello-Smith, W. M., W. Jiang, H. Zhang, A. D. Ozkan, Y. C. Lin, C. N. Pham, J. J. Lacroix and Y. Luo (2019). "A mechanism for the activation of the mechanosensitive Piezo1 channel by the small molecule Yoda1." *Nat Commun* **10**(1): 4503.
- Bower, A. F. (2009). *Applied Mechanics of Solids*. Boca Raton, CRC Press.
- Colom, A., E. Derivery, S. Soleimanpour, C. Tomba, M. D. Molin, N. Sakai, M. Gonzalez-Gaitan, S. Matile and A. Roux (2018). "A fluorescent membrane tension probe." *Nat Chem* **10**(11): 1118-1125.
- Cox, C. D., C. Bae, L. Ziegler, S. Hartley, V. Nikolova-Krstevski, P. R. Rohde, C. A. Ng, F. Sachs, P. A. Gottlieb and B. Martinac (2016). "Removal of the mechanoprotective influence of the cytoskeleton reveals PIEZO1 is gated by bilayer tension." *Nat Commun* **7**: 10366.
- Cox, C. D., N. Bavi and B. Martinac (2019). "Biophysical Principles of Ion-Channel-Mediated Mechanosensory Transduction." *Cell Rep* **29**(1): 1-12.
- Fillafer, C., M. Mussel, J. Muchowski and M. F. Schneider (2018). "Cell Surface Deformation during an Action Potential." *Biophys J* **114**(2): 410-418.
- Golani, G., N. Ariotti, R. G. Parton and M. M. Kozlov (2019). "Membrane Curvature and Tension Control the Formation and Collapse of Caveolar Superstructures." *Dev Cell* **48**(4): 523-538 e524.
- Haselwandter, C. A. and R. MacKinnon (2018). "Piezo's membrane footprint and its contribution to mechanosensitivity." *Elife* **7**.
- Kuchel, P. W., C. D. Cox, D. Daners, D. Shishmarev and P. Galvosas (2021). "Surface model of the human red blood cell simulating changes in membrane curvature under strain." *Sci Rep* **11**(1): 13712.
- Lewis, A. H. and J. Grandl (2015). "Mechanical sensitivity of Piezo1 ion channels can be tuned by cellular membrane tension." *Elife* **4**.
- Lewis, A. H. and J. Grandl (2021). "Piezo1 ion channels inherently function as independent mechanotransducers." *Elife* **10**.
- Luo, T., K. Mohan, P. A. Iglesias and D. N. Robinson (2013). "Molecular mechanisms of cellular mechanosensing." *Nat Mater* **12**(11): 1064-1071.
- Rangamani, P., K. K. Mandadap and G. Oster (2014). "Protein-induced membrane curvature alters local membrane tension." *Biophys J* **107**(3): 751-762.
- Ridone, P., S. L. Grage, A. Patkunarajah, A. R. Battle, A. S. Ulrich and B. Martinac (2018). "Force-from-lipids gating of mechanosensitive channels modulated by PUFAs." *J Mech Behav Biomed Mater* **79**: 158-167.
- Shi, Z. and T. Baumgart (2015). "Membrane tension and peripheral protein density mediate membrane shape transitions." *Nat Commun* **6**: 5974.
- Suchyna, T. M., V. S. Markin and F. Sachs (2009). "Biophysics and structure of the patch and the gigaseal." *Biophys J* **97**(3): 738-747.
- Syeda, R., J. Xu, A. E. Dubin, B. Coste, J. Mathur, T. Huynh, J. Matzen, J. Lao, D. C. Tully, I. H. Engels, H. M. Petrassi, A. M. Schumacher, M. Montal, M. Bandell and A. Patapoutian (2015). "Chemical activation of the mechanotransduction channel Piezo1." *Elife* **4**.
- Vaisey, G., P. Banerjee, A. J. North, C. A. Haselwandter and R. MacKinnon (2022). "Piezo1 as a force-through-membrane sensor in red blood cells." *Elife* **11**.
- Wang, Y., S. Chi, H. Guo, G. Li, L. Wang, Q. Zhao, Y. Rao, L. Zu, W. He and B. Xiao (2018). "A lever-like transduction pathway for long-distance chemical- and mechano-gating of the mechanosensitive Piezo1 channel." *Nat Commun* **9**(1): 1300.
- Yao, M., A. Tijore, D. Cheng, J. V. Li, A. Hariharan, B. Martinac, G. Tran Van Nhieu, C. D. Cox and M. Sheetz (2022). "Force- and cell state-dependent recruitment of Piezo1 drives focal adhesion dynamics and calcium entry." *Sci Adv* **8**(45): eabo1461.
- Young, M., A. H. Lewis and J. Grandl (2022). "Physics of mechanotransduction by Piezo ion channels." *J Gen Physiol* **154**(7).

REVIEWER COMMENTS

Reviewer #1 (Remarks to the Author):

The authors have made an effort to address the points raised in my previous report. However, the central criticisms detailed in my previous report---namely, the inconclusive nature of the comparisons between model and experiments, the many degrees of freedom in the model, and the apparently flawed physical picture underlying the model and associated interpretation of experimental data---still stand. Unfortunately, this paper therefore does not add much in terms of new understanding.

The authors' statement in their response that "local membrane geometry at the nanometer scale and its effect on PIEZO1 gating will not be further explored in this study" (and associated/preceding discussion) is not satisfactory, since it is precisely that nanometer geometry (and membrane tension) that will be crucial for the PIEZO1 gating (and, potentially, PIEZO1 localization) the authors aim to study here. Furthermore, the discussion of membrane tension in the manuscript is misleading: The manuscript creates the impression that somehow membrane tension is decoupled from membrane shape, which is not correct---over the past 50 years (going back to the seminal work of Helfrich et al.), a whole field of research has dealt with the membrane mechanics of that coupling, both from the perspective of experiment and theory.

Having said this, the experiments described in this paper are of some interest, although it is not clear what they might mean from the perspective of physics or biology. It is unfortunate that the authors did not manage to come up with a simple physical model (or several simple models) that allow definite predictions and comparisons with experiments, thus producing an understanding of the data described here and insight into what the more general meaning of this data for PIEZO1's biological function might be. I leave it up to the editors to decide whether such a paper---on an interesting general topic with some interesting data, somewhat superficial content, and little in terms of solid new understanding---is suitable for publication.

Reviewer #2 (Remarks to the Author):

The authors provide an updated version of the manuscript with compelling and state-of-the-art new data (e.g. Flipr imaging) that further supports their conclusions. Moreover, the updated text makes easier to understand some complex findings. All in all, this updated manuscript addresses my concerns and I firmly support its publication. Congratulations to the authors

Minor comments:

- Lines 187-203, p9. Check the whole paragraph for fluidity and spelling. Many "the" seem displaced/missed from sentence re-writing.

- Check calcium and Ca²⁺ (and superindex) for consistency

- Regarding minor comment 2 in the previous revision round: I agree delving on how Piezo1 participates in F-actin accumulation (membrane mechanics vs calcium signalling) is out of the scope of this paper, and maybe my "actin buckle formation requires Piezo1" was an overstatement. However, fig.S8a shows Piezo1KO strongly prevents F-actin accumulation, which to me suggests the channel is indeed required.

- Related to the previous point: line 293-297, p14 'We uncovered a clear trend that a larger tip angle facilitated a more pronounced F-actin accumulation in the neck region during aspiration [...] in all PIEZO1 genotypes (Fig. S8a)". As already stated, In the Piezo1 KO genotype there is a strong reduction of F-actin accumulation. The authors should at least mention this. Otherwise, the sentence is not accurate.

Reviewer #3 (Remarks to the Author):

The author describes a fluorescent micropipette aspiration assay for simultaneous visualization of intracellular calcium dynamics and cytoskeletal architecture in real-time and FLIM analysis was used. The idea is novel and the findings lead to future biomedical applications.

Minor comments-

Authors have shown F-actin mobility modulates PIEZO1 activation at aspirated cell tongue where HEK293T cells are used. How this analysis correlates with the local membrane tension as shown for RBC based results ?

What are the excitation and emission wavelengths filters used for fluorescence signal measurement ?

What was the acquisition time for single FLIM image ?

What are the challenges of FLIM measurement in this study ?

Point-by-point responses to the reviewers' comments

Our point-by-point responses are denoted in **blue**, while the corresponding actions are detailed in **purple**. Changes within the revised manuscript are marked in **red** for easy identification.

REVIEWER COMMENTS

Response to all reviewers:

Response: We noticed all reviewers raised the uncertainty about the interaction of F-actin and PIEZO1 during micropipette aspiration. We appreciate the critiques which help us further improve the quality of our work.

Action taken: To address the request, we conducted additional fluorescent micropipette aspiration assays (fMPA) on hP1-mCherry-1591 HEK293T to monitor the PIEZO1 movement and co-localization with F-actin. The results are presented in **Fig. R1** (Fig. 7 in the revised manuscript) and incorporated into the updated schematic illustrated in **Fig. R2** (Fig. 8 in the revised manuscript).

Fig. 7. PIEZO1 co-localizes with F-actin during micropipette aspiration in HEK293T cells.

a and b Confocal images of hP1-mCherry-1591 HEK293T cell aspirated at $\Delta p = -25$ mmHg by micropipette ($\theta = 10^\circ$). The F-actin (cyan, 1st row) and Piezo1 (magenta, 2nd row) dynamics

were monitored during aspiration using a spinning disk confocal microscope. Both adhered (a) and suspended (b) HEK293T cell showed PIEZO1 clusters (*white arrow*) movement from the cell body to tongue during aspiration. A symmetric axis was drawn along the aspirating micropipette and across the cell body. Taking the fore-end of the cell body (i.e., left edge) as $x = 0$, mean PIEZO1 intensity were plotted across the adhered HEK293 (c) in a and suspended one (d) in b, respectively. (e) PIEZO and F-actin co-localization was quantified during micropipette aspiration ($\theta = 10^\circ$). Results were analyzed from $n \geq 8$ cell from two independent experiments. (f) Suspended HEK293T cells had higher co-localization of PIEZO1/F-actin at both resting ($t = 0s$) and aspirated time points ($t = 5, 20s$) compared to aspirated HEK293T cells. Images demonstrate a concurrent movement of F-actin and PIEZO1 from the cell body to the cell tongue during aspiration in suspended HEK293T cells. Results were compared by unpaired, two-tailed Student's t -test. * = $p < 0.05$; **** = $p < 0.0001$.

We have also expanded our Results section on the role of F-actin dynamics and network structure in modulating PIEZO1 activity.

Page 20 Line 25 - Page 21 Line 13:

“To confirm that the lower Ca^{2+} mobilization in adhered compared to suspended HEK293T cells was not due to decreased PIEZO1 movement not the micropipette, adhered (Fig. 7a) and suspended (Fig. 7b) hP1-mCherry-1591 HEK293T were aspirated at $\Delta p = -25$ mmHg under all angle micropipettes (i.e., $\theta = 0^\circ, 5^\circ, \text{ and } 10^\circ$) to monitor the PIEZO1 movement on a spinning disk confocal microscope. PIEZO1 in both adhered and suspended responded to the aspiration with high signal clusters (*white arrow*) moving into the micropipette, noted by a right shifting average PIEZO1 intensity overtime along the micropipette symmetric axis (Fig. 7c, *adhered*; Fig. 7d, *suspended*). To verify whether PIEZO1 movement was related to the F-actin movement, we analyzed the co-localization factor¹ between PIEZO1 and F-actin. Results showed that the co-localization significantly increased when the suspended HEK293T cell was aspirated (Fig. 7e, *blue*), indicating that PIEZO1 and F-actin responded and moved into the micropipette at the same time until $t = 5$ s. Over the time, the co-localization dropped after $t > 5$ s. Rather than accumulating with the F-actin at the neck of the micropipette (Fig. 4a), PIEZO1 moves with the lipid bilayer into the micropipette. On the other hand, when the cell was adhered to the FN surface, the co-localization was initially lower when compared to the suspended cells, indicating that transformed cell lines do not recruit PIEZO1 into focal adhesions². Meanwhile, when the adhered cell was aspirated, the co-localization factor had minor changes over time,

further suggesting that PIEZO1 movement occurs independently of F-actin in response to aspiration. Such results were consistent under aspiration with different θ micropipettes (Fig. S11). Hence, PIEZO1 movement is solely driven by the aspiration force and has limited interaction with F-actin dynamics in this model.”

Fig. R2 (Fig. 8 in revised manuscript). Microgeometry and F-actin interplay during fMPA modulate PIEZO1 activity at the tongue of aspirated cells. Microgeometry parameters, such as tip angle θ and opening diameter d , globally enhance PIEZO1 activity on the cell membrane while F-actin filaments inside the cell concurrently respond to mechanical aspiration. Reorganized F-actin accumulates at the neck of the aspirated cell, acting as a physical constriction on the membrane to isolate the propagation of membrane tension from the tongue. Concentrated tension at the tongue induces PIEZO1 hyperactivity and results in strong calcium mobilization. Hinderance of F-actin accumulation at the neck, due to either lack of physical constriction (micropipette tip angle $\theta = 0^\circ$) or prohibited actin mobility when cells are adhered to the extracellular matrix, results in low PIEZO1 activity at the tongue. Figure 8 was created with BioRender.com and released under a Creative Commons Attribution-NonCommercial-NoDerivs 4.0 International license <https://creativecommons.org/licenses/by-nc-nd/4.0/deed.en>.

Reviewer #1 (Remarks to the Author):

The authors have made an effort to address the points raised in my previous report. However, the central criticisms detailed in my previous report---namely, the inconclusive nature of the comparisons between model and experiments, the many degrees of freedom in the model, and the apparently flawed physical picture underlying the model and associated interpretation of experimental data---still stand. Unfortunately, this paper therefore does not add much in terms of new understanding.

Response: We appreciate the reviewer's concerns and have made significant improvements to address these issues. We have optimized the tension simulation in the Results section 2.5 (originally Fig. S9a and b) to better demonstrate the effect of F-actin accumulation on membrane tension in aspirated HEK293T cells. Our improved modeling now clearly illustrates that F-actin accumulation at the neck region leads to a significant increase in membrane tension at the neck and dome of the cell tongue (Fig. R3), which correlates with the higher PIEZO1 activities observed in the fMPA assay (Fig. 4a and c).

Fig. R3. Illustration of F-actin constriction on aspirated cell neck with FEA simulation. a and b. Normalized maximum principal stress distribution simulation of an aspirated HEK293T cell. The cell was either accumulation-free (a, (-) Accumulation), or with F-actin accumulation at the neck (b, (+) Accumulation). Maximum principal stress values were chosen to represent the membrane tension under either condition and were normalized based on the maximum value of both conditions.

Action taken:

Page 34 Line 25 - Page 35 Line 2:

“When the F-actin accumulates at the neck region of aspirated HEK293T cell, the material properties are expected to significantly increase^{3,4}. In our proposed material model, we introduce an adjustable parameter α (Eq. 11) which requires calibration to align with the experimental results:

$$T_{\text{total}} = t_{\text{aspiration}} + t_{\text{acc}} \quad (10)$$

$$E_{\text{acc}} = E_{\text{aspiration}} + (\alpha - 1) \times E_{\text{aspiration}} \times H(t - t_{\text{aspiration}}) \quad (11)$$

Where $t_{\text{aspiration}}$ is the time point cell get aspirated into the micropipette and H is the Heaviside function.

Fig. S10 in the original manuscript (Fig. R1) was now removed and merged with Fig. 5 (g and h) in the revised manuscript.

The authors' statement in their response that "local membrane geometry at the nanometer scale and its effect on PIEZO1 gating will not be further explored in this study" (and associated/preceding discussion) is not satisfactory, since it is precisely that nanometer geometry (and membrane tension) that will be crucial for the PIEZO1 gating (and, potentially, PIEZO1 localization) the authors aim to study here. Furthermore, the discussion of membrane tension in the manuscript is misleading: The manuscript creates the impression that somehow membrane tension is decoupled from membrane shape, which is not correct---over the past 50 years (going back to the seminal work of Helfrich et al.), a whole field of research has dealt with the membrane mechanics of that coupling, both from the perspective of experiment and theory.

Having said this, the experiments described in this paper are of some interest, although it is not clear what they might mean from the perspective of physics or biology. It is unfortunate that the authors did not manage to come up with a simple physical model (or several simple models) that allow definite predictions and comparisons with experiments, thus producing an understanding of the data described here and insight into what the more general meaning of this data for PIEZO1's biological function might be. I leave it up to the editors to decide whether such a paper---on an interesting general topic with some interesting data, somewhat superficial content, and little in terms of solid new understanding---is suitable for publication.

Response: We appreciate the reviewer's perspective on the importance of nanoscale membrane geometry and tension in PIEZO1 gating and localization. Force-stretched lipid bilayers may also flatten the PIEZO1 footprint and favor PIEZO1 gating⁵⁻⁸. In our study, we utilized two elegant PIEZO1-expressing cell systems with distinct nanoscale membrane curvatures around

PIEZO1: primary suspending RBCs with smooth and flat membranes and transformed adherent HEK293T cells with invaginated membranes (Fig. R4).

Fig. R4. Schematic of distinct membrane curvatures around PIEZO1 in RBC (a; **smooth and flat**) and HEK293T (b; **invaginated**), respectively.

Due to the unique cytoskeletal structure and high turgor pressure of RBCs, local membrane curvature varies little along the surface at isotonic conditions⁹⁻¹¹. Thus, PIEZO1 does not tend to form clusters in RBCs (Fig. R4a). During aspiration, there will also be limited curvature change to the membrane around PIEZO1 channels. On the other hand, large negative curvature in the membrane of HEK293T cells leads to forming PIEZO1 clusters in the invagination (Fig.R4b)¹². Figure R4 Panels a-b were created with BioRender.com released under a Creative Commons Attribution-NonCommercial-NoDerivs 4.0 International license <https://creativecommons.org/licenses/by-nc-nd/4.0/deed.en>.

Interestingly, despite the distinct nanoscale membrane curvatures and PIEZO1 localization patterns in these two cell systems, both cell lines exhibited similar trends in PIEZO1-mediated Ca^{2+} response under microscale geometrical modulation of the aspirating micropipettes. This suggests that the observed phenomenon in our fMPA systems is largely independent of nanoscale curvatures. Furthermore, most aspirated HEK293T cells did not show significant Ca^{2+} response until seconds after large deformation (i.e., after F-actin accumulation, Fig. S8d in the original manuscript). In this case, the large deformation in the aspirated tongue should have already flattened the invaginations and caused significant increase in the microscale membrane curvature, yet limited Ca^{2+} influx was noticed. Thus, we interpret the later PIEZO1 gating in our results to be primarily driven by the further increased lipid stretching force which eventually open the mechanosensitive ion channels by lateral stretching and nanoscale footprint unbending.

Action taken: We have revised the discussion to clarify that our findings suggest a dominant role of microscale geometrical modulation in PIEZO1 gating, which appears to be largely independent of nanoscale membrane curvature:

Page 23 Line 10 - Line 22: "Our study utilized two PIEZO1-expressing cell systems with distinct nanoscale membrane curvatures: RBCs with smooth and flat membranes and HEK293T cells with invaginated membranes. Despite these differences in nanoscale geometry and PIEZO1 localization, both cell lines exhibited similar trends in PIEZO1-mediated Ca^{2+} response under microscale geometrical modulation of the aspirating micropipettes. This suggests that the observed phenomenon is largely independent of microscale curvatures. Furthermore, the limited Ca^{2+} influx in HEK293T cells during the initial aspiration phase (Fig. S8d), when invaginations are expected to flatten, indicates that the later PIEZO1 gating in our system is primarily driven by force-from-lipid mechanisms or lateral force-induced nanoscale footprint changes^{5-8,13}, rather than local microscale membrane curvature alterations^{5,6,14}. However, it is important to consider the complex interplay between membrane tension¹⁵, curvature, and PIEZO1 gating in future studies aimed at understanding the effect of microgeometry on single-cell mechanosensing."

Reviewer #2 (Remarks to the Author):

The authors provide an updated version of the manuscript with compelling and state-of-the-art new data (e.g. Flipr imaging) that further supports their conclusions. Moreover, the updated text makes easier to understand some complex findings. All in all, this updated manuscript addresses my concerns and I firmly support its publication. Congratulations to the authors

Minor comments:

- Lines 187-203, p9. Check the whole paragraph for fluidity and spelling. Many "the" seem displaced/missed from sentence re-writing.

Response: We thank the reviewer for raising the mistake.

Action taken: The fluidity and spelling in the paragraph has been checked and corrected.

- Check calcium and Ca²⁺ (and superindex) for consistency

Action taken: The consistency has been checked and complied.

- Regarding minor comment 2 in the previous revision round: I agree delving on how Piezo1 participates in F-actin accumulation (membrane mechanics vs calcium signalling) is out of the scope of this paper, and maybe my "actin buckle formation requires Piezo1" was an overstatement. However, fig.S8a shows Piezo1KO strongly prevents F-actin accumulation, which to me suggests the channel is indeed required.

Action taken:

Page 14 Line 9: To avoid overstatement and misleading readers, the sentence has been rephrased "...a higher PIEZO1 expression upregulated the probability of observing accumulation..."

- Related to the previous point: line 293-297, p14 "We uncovered a clear trend that a larger tip angle facilitated a more pronounced F-actin accumulation in the neck region during aspiration [...] in all PIEZO1 genotypes (Fig. S8a)". As already stated, In the Piezo1 KO genotype there is a strong reduction of F-actin accumulation. The authors should at least mention this. Otherwise, the sentence is not accurate.

Action taken: the statement has been updated

Page 14 Line 15 - Line 21: “We uncovered a clear trend that a larger tip angle facilitated a more pronounced F-actin accumulation ... in PIEZO1-WT and PIEZO1-OE genotypes (Fig. S8a) whereas absence of the mechanosensitive ion channel in PIEZO1-KO genotype leads to few accumulation events recorded in this study and the trend is not clear. Nevertheless, results suggest that...”

Reviewer #3 (Remarks to the Author):

The author describes a fluorescent micropipette aspiration assay for simultaneous visualization of intracellular calcium dynamics and cytoskeletal architecture in real-time and FLIM analysis was used. The idea is novel and the findings lead to future biomedical applications.

Minor comments-

Authors have shown F-actin mobility modulates PIEZO1 activation at aspirated cell tongue where HEK293T cells are used. How this analysis correlates with the local membrane tension as shown for RBC based results?

Response: The FLIM measurement on RBC proves the microscale geometrical modulation on local membrane tension, showing that higher tip angles (i.e., $\theta = 10^\circ$) would facilitate a higher membrane tension at the aspirated tongue. Meanwhile, this higher tension was further amplified by the physical constriction after F-actin accumulating at the aspirated neck region. Thereby, our results demonstrated a further strengthened PIEZO1 activities at the tongue.

What are the excitation and emission wavelengths filters used for fluorescence signal measurement ?

Action taken: Information has been added to the Method section:

Page 32 Line 14 - Line 15: “A filter cube with 475/28 nm excitation filter, 561 nm dichroic cut-off mirror, and 609/57 nm emission filter was in place to separate fluorescence.”

What was the acquisition time for single FLIM image ?

Action taken: Information has been added

Page 32 Line 11 - Line 13: “To ensure enough photon counting for lifetime analysis, temporal resolution was sacrificed and a total number of 5 frames (total acquisition time = 20 s) were stacked for each acquisition.”

What are the challenges of FLIM measurement in this study?

Response: There are two major challenges in FLIM measurements: \

1. Temporal resolution. Since the photon count of the measurement is limited within a single acquisition, temporal resolution was sacrificed to ensure that photon counts were sufficient for FLIM analysis.
2. The FLIM measurements on Flipper-TR dye probes also depend on the lipid composition. Therefore, it is hard to perform calibration to convert the lifetime to tension magnitude.

Action taken: the challenges were added to mention in the Method with the statement of acquisition time.

Bibliography

- 1 Adler, J., Pagakis, S. N. & Parmryd, I. Replicate-based noise corrected correlation for accurate measurements of colocalization. *J Microsc* **230**, 121-133 (2008). <https://doi.org:10.1111/j.1365-2818.2008.01967.x>
- 2 Yao, M. *et al.* Force- and cell state-dependent recruitment of Piezo1 drives focal adhesion dynamics and calcium entry. *Sci Adv* **8**, eabo1461 (2022). <https://doi.org:10.1126/sciadv.abo1461>
- 3 Bhadriraju, K. & Hansen, L. K. Extracellular matrix- and cytoskeleton-dependent changes in cell shape and stiffness. *Exp Cell Res* **278**, 92-100 (2002). <https://doi.org:10.1006/excr.2002.5557>
- 4 Tavares, S. *et al.* Actin stress fiber organization promotes cell stiffening and proliferation of pre-invasive breast cancer cells. *Nat Commun* **8**, 15237 (2017). <https://doi.org:10.1038/ncomms15237>
- 5 Haselwandter, C. A., Guo, Y. R., Fu, Z. & MacKinnon, R. Quantitative prediction and measurement of Piezo's membrane footprint. *Proc Natl Acad Sci U S A* **119**, e2208027119 (2022). <https://doi.org:10.1073/pnas.2208027119>
- 6 Haselwandter, C. A. & MacKinnon, R. Piezo's membrane footprint and its contribution to mechanosensitivity. *Elife* **7** (2018). <https://doi.org:10.7554/eLife.41968>
- 7 Guo, Y. R. & MacKinnon, R. Structure-based membrane dome mechanism for Piezo mechanosensitivity. *Elife* **6** (2017). <https://doi.org:10.7554/eLife.33660>
- 8 Haselwandter, C. A., Guo, Y. R., Fu, Z. & MacKinnon, R. Elastic properties and shape of the Piezo dome underlying its mechanosensory function. *Proc Natl Acad Sci U S A* **119**, e2208034119 (2022). <https://doi.org:10.1073/pnas.2208034119>
- 9 Vaisey, G., Banerjee, P., North, A. J., Haselwandter, C. A. & MacKinnon, R. Piezo1 as a force-through-membrane sensor in red blood cells. *Elife* **11** (2022). <https://doi.org:10.7554/eLife.82621>
- 10 Li, H. & Lykotrafitis, G. Erythrocyte membrane model with explicit description of the lipid bilayer and the spectrin network. *Biophys J* **107**, 642-653 (2014). <https://doi.org:10.1016/j.bpj.2014.06.031>
- 11 Buys, A. V. *et al.* Changes in red blood cell membrane structure in type 2 diabetes: a scanning electron and atomic force microscopy study. *Cardiovasc Diabetol* **12**, 25 (2013). <https://doi.org:10.1186/1475-2840-12-25>
- 12 Yang, S. *et al.* Membrane curvature governs the distribution of Piezo1 in live cells. *Nat Commun* **13**, 7467 (2022). <https://doi.org:10.1038/s41467-022-35034-6>
- 13 Young, M., Lewis, A. H. & Grandl, J. Physics of mechanotransduction by Piezo ion channels. *J Gen Physiol* **154** (2022). <https://doi.org:10.1085/jgp.202113044>
- 14 Lewis, A. H. & Grandl, J. Mechanical sensitivity of Piezo1 ion channels can be tuned by cellular membrane tension. *Elife* **4** (2015). <https://doi.org:10.7554/eLife.12088>
- 15 Bavi, N., Cox, C. D., Nikolaev, Y. A. & Martinac, B. Molecular insights into the force-from-lipids gating of mechanosensitive channels. *Current Opinion in Physiology*, 100706 (2023).

REVIEWERS' COMMENTS

Reviewer #3 (Remarks to the Author):

The author addressed all the comments raised. The inclusion of new results and discussion strengthen the manuscript.

Point-by-point responses to the reviewers' comments

Our point-by-point responses are denoted in blue, while the corresponding actions are detailed in purple.

REVIEWER COMMENTS

Reviewer #3 (Remarks to the Author):

The author addressed all the comments raised. The inclusion of new results and discussion strengthen the manuscript.

Response: We much appreciated the reviewer's positive comment!